# mTOR contributes to endothelium-dependent vasorelaxation by promoting eNOS expression and preventing eNOS uncoupling

Yiying Wang[1,2], Qiannan Li[1,2], Zhiyang Zhang[1,2], Kai Peng[1,2], Dai-Min Zhang [3,4], Qianlu Yang[1,2], Anthony G. Passerini [5], Scott I. Simon [5] & ChongXiu Sun [1,2✉]

Clinically used inhibitors of mammalian target of rapamycin (mTOR) negatively impacts endothelial-dependent vasodilatation (EDD) through unidentified mechanisms. Here we show that either the endothelium-specific deletion of *Mtor* to inhibit both mTOR complexes, or depletion of Raptor or Rictor to disrupt mTORC1 or mTORC2, causes impaired EDD, accompanied by reduced NO in the serum of mice. Consistently, inhibition of mTOR decreases NO production by human and mouse EC. Specifically, inhibition of mTORC1 suppresses *eNOS* gene expression, due to impairment in p70S6K-mediated post-transcriptional regulation of the transcription factor KLF2 expression. In contrast to mTORC1 inhibition, a positive-feedback between MAPK (p38 and JNK) activation and Nox2 upregulation contributes to the excessive generation of reactive oxygen species (ROS), which causes eNOS uncoupling and decreased NO bioavailability in mTORC2-inhibited EC. Adeno-associated virus-mediated EC-specific overexpression of KLF2 or suppression of Nox2 restores EDD function in endothelial mTORC1- or mTORC2-inhibited mice.

[1] Key Laboratory of Targeted Intervention of Cardiovascular Disease, Collaborative Innovation Center for Cardiovascular Disease Translational Medicine, Nanjing Medical University, Nanjing, China. [2] Key laboratory of Human Functional Genomics of Jiangsu Province, Nanjing, China. [3] Department of Cardiology, Nanjing First Hospital, Nanjing Medical University, Nanjing, China. [4] Department of Cardiology, Sir Run Run Hospital, Nanjing Medical University, Nanjing, China. [5] Department of Biomedical Engineering, University of California Davis, Davis, CA, USA. ✉email: cxsun@njmu.edu.cn

Coronary artery disease (CAD), manifests by narrowing and hardening of the coronary artery and, ultimately, the compromised blood supply to the myocardium, ranks as the leading cause of mortality worldwide. Drug-eluting stents (DES) have proven to be an effective intervention to restore lumen lost to CAD. Compared with traditional bare-metal stents (BMS)[1], DES reduces in-stent restenosis through the local release of anti-proliferative agents to inhibit neointima formation. However, the use of DES in some patients is associated with a risk for adverse effects such as delayed arterial healing and poor re-endothelialization. Moreover, numerous reports have linked DES with impaired endothelial-dependent arterial dilatation both proximal and distal to the interventional segment as late as 6 months post-implantation[2,3].

Currently, the majority of DES elute rapamycin (also known as sirolimus) or its analogs such as everolimus, which are inhibitors of the mammalian target of rapamycin (mTOR), an atypical serine/threonine-protein kinase. mTOR forms two functionally distinct complexes, mTOR complex 1 (mTORC1) and mTOR complex 2 (mTORC2), both of which contain mTOR and mLST8. Raptor and Rictor are unique components of mTORC1 and mTORC2, respectively. mTORC1 phosphorylates its substrates S6 kinase-1 (S6K1) and eukaryotic translation initiation factor 4E (eIF4E)-binding protein-1 (4EBP-1) to regulate protein translation, cell growth, and proliferation. mTORC2 functions as the major kinase for protein kinase B (AKT) and protein kinase Cα (PKCα), controlling cell survival and cytoskeleton organization[4].

Rapamycin, one of the first-generation mTOR allosteric inhibitors, has been widely used for preventing transplant rejection based on its immunosuppressive effects. It acts in an anti-proliferative capacity to inhibit in-stent arterial restenosis, by promoting cell-cycle arrest at the late G1 phase[5]. The selective binding of rapamycin to FK-506-binding protein 12 (FKBP12) caused acute disassociation of Raptor from mTOR, leading to highly sensitive and specific mTORC1 inhibition[6]. However, prolonged treatment of rapamycin was also reported to cause the disassembly of mTORC2[7]. Recently, second-generation mTOR kinase inhibitors such as torin 1 have been developed. Potent in suppressing both mTOR complexes by competing with ATP for binding to the kinase, they have been investigated as anticancer agents[8].

Endothelial cells (EC), located at the interface between the vessel wall and blood, play a key role in the local control of vasomotor tone mainly through synthesizing and releasing endothelium-derived relaxing factors (EDRF) such as nitric oxide (NO)[9]. Endothelial NO synthase (eNOS), the predominant NOS isoform expressed in EC and within the vasculature, is responsible for most of the NO production from l-arginine and contributes essentially to the maintenance of vascular function and homeostasis. When NO diffuses to the sub-endothelial matrix, it binds and activates soluble guanylate cyclase on the membrane of smooth muscle cells (SMC), eliciting endothelium-dependent relaxation by catalyzing the production of cyclic guanosine monophosphate (cGMP) and activation of the downstream protein kinase G (PKG) in SMC[10].

In addition to being subject to transcriptional and translational regulation of gene expression, the activity of eNOS can be modulated post-translationally[11]. For example, eNOS undergoes phosphorylation on its serine (Ser) and threonine (Thr) residues. Phosphorylated Ser1177 and Ser635 were associated with an active conformation while phosphorylation at Thr495 and Ser116 promoted inhibition. Moreover, the bioactivity of NO can be further regulated by its ready interaction with superoxide anion ($O_2^{-\cdot}$), one kind of reactive oxygen species (ROS), to form peroxynitrite ($ONOO^-$). $ONOO^-$ is known to oxidize $(6R)-5,6,7,8$-tetrahydro-L-biopterin ($BH_4$), the cofactor of eNOS, transforming it to dysfunctional $BH_2$, which causes eNOS to catalyze $O_2$ reduction uncoupled from NO production. In this manner, eNOS can be converted from a bioactive NO-producing enzyme to one that generates deleterious $O_2^-$, a process which has been referred to as eNOS uncoupling[12,13]. Risk factors for cardiovascular disease are known to lead to enhanced ROS production in the vessel wall[13].

A detrimental role for mTOR inhibition in promoting vasomotor dysfunction has also been documented in animal studies[14,15]. For example, infusion of mice with rapamycin increased systolic blood pressure through a mechanism involving an increase in eNOS phosphorylation at Thr495[15]. Furthermore, inactivation of mTORC2 in perivascular adipose tissue impaired endothelium-dependent relaxation through upregulation of inflammatory mediators[14]. Despite these observations, the mechanisms underlying the putative impairment of the eNOS/NO pathway by mTOR inhibition as relates to the complications of DES are ill-defined.

Here we investigated the mechanism by which mTOR signaling exerts its effect on eNOS production and function in arterial EC to affect NO bioavailability and vasomotor function. We proposed that inhibition of mTOR signaling through mTORC1 and mTORC2 could affect complementary mechanisms to impair endothelial-mediated dilatation. We demonstrate that EC-specific *Mtor-, Rptor-,* or *Rictor*-knockout mice displayed impaired EC-dependent aortic vasodilation in response to increased hemodynamic shear stress or acetylcholine (ACh), accompanied by decreased NO in the serum. Additional studies in primary human arterial EC (HAEC) revealed that both mTORC1 and mTORC2 contributed to the regulation of NO bioavailability. Mechanistically, mTORC1/p70S6K was required for KLF2-mediated *eNOS* expression, while intact mTORC2 prevented NADPH oxidase 2 (Nox2)/ROS-dependent eNOS uncoupling. Lastly, the potential therapeutic benefit of modulation of KLF2 and Nox2 in preserving EC-dependent vasodilation was confirmed in endothelial *Rptor-* or *Rictor*-knockout mice.

## Results

**EC-specific deletion of *Mtor, Rptor,* or *Rictor* caused impaired arterial relaxation.** Flow-mediated dilation (FMD) is the vasodilatory response of the artery to increased hemodynamic shear stress. It reflects the ability of EC to produce NO and is a well-recognized indicator of EC-dependent vascular function[16]. As a NO donor, GTN elicits relaxation independent of EC. In this study, ultrasound imaging showed that either FMD or GTN caused an increase of 15–25% in the maximum systolic diameter of the WT mouse abdominal aorta (Supplementary Fig. 1a).

Consistent with clinical observations that mTOR inhibition impairs endothelial-mediated vasomotor function, i.p. injection of mice with rapamycin at a dose that led to a comparable blood concentration of rapamycin to that in patients implanted with DES[17], caused a decrease of 31.9% in EC-dependent dilation (EDD) but not EC-independent dilation (EID) (Supplementary Fig. 1b).

To further define the role of mTOR in the regulation of EC-dependent vascular tone, tamoxifen-inducible EC-specific *Mtor* knockout ($Mtor^{EC-/-}$) mice were generated that inhibit both mTOR complexes in EC. Furthermore, EC-specific *Rptor* (encoding Raptor) or Rictor-knockout ($Rptor^{EC-/-}$ or $Rictor^{EC-/-}$) mice were established to specifically disrupt the endothelial mTORC1 or mTORC2[18].

Deficiency of mTOR, Raptor, or Rictor in EC decreased FMD by 33.9%, 40.1%, and 41.1%, respectively compared to the wild-type littermates (Fig. 1a–c). In contrast, relaxation caused by GTN (Supplementary Fig. 1d, e) as well as other measurements such as heart rate (Supplementary Fig. 1f–h) were not affected by any gene knockout.

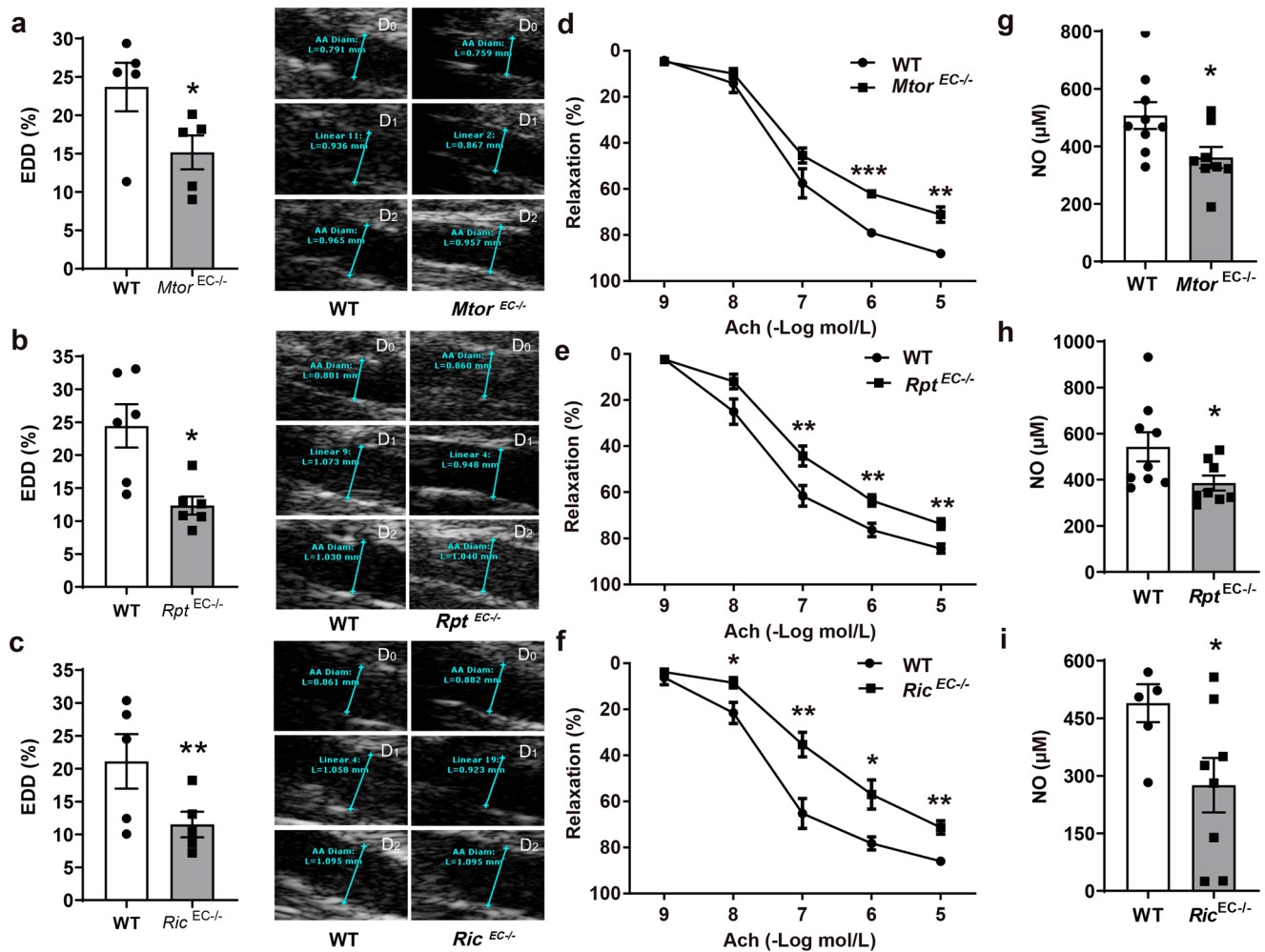

**Fig. 1 EC-specific deletion of *Mtor*, *Rptor*, or *Rictor* caused impaired relaxation of the artery accompanying decreased NO bioavailability.**
**a–c** Endothelium-dependent vasodilation (EDD) and endothelium-independent vasodilation (EID) of the abdominal aortas of $Mtor^{EC-/-}$, $Rptor$ $(Rpt)^{EC-/-}$, $Rictor$ $(Ric)^{EC-/-}$ mice and their wild type (WT) littermates were measured by transcutaneous ultrasound imaging ($n = 5$-6) and calculated as follows: EDD% = $(D_1-D_0)/D_0 \times 100$; EID% = $(D_2-D_0)/D_0 \times 100$. Where $D_0$ is the mean diameter of the murine abdominal aorta over three cardiac cycles at baseline, $D_1$ is the maximum systolic diameter after flow-mediated dilation, and $D_2$ is the maximum systolic diameter after nitroglycerin infusion. Shown to the right are representative images of the abdominal aortas. Scale = 0.5 mm. **d–f** The dose responses of aortic rings to $10^{-9}$-$10^{-5}$ mol/L acetylcholine (Ach) were evaluated by Myograph ($n = 5$-8). **g–i** NO levels in the mouse serum was measured with a colorimetric reaction ($n = 7$-9). Error bars correspond to standard error of the mean (SEM). $*p < 0.05$; $**p < 0.01$; $***p < 0.001$ vs. WT littermates; two-tailed unpaired $t$ test.

These in vivo findings of decreased FMD were consistent with ex vivo studies which indicated that aortic rings isolated from $Mtor^{EC-/-}$, $Rptor^{EC-/-}$ or $Rictor^{EC-/-}$ mice had an impaired EC-dependent vasodilatory response to ACh, a NO-dependent vasodilator (Fig. 1d–f). It is noteworthy that norepinephrine (NE)-induced contraction was also enhanced by EC deficiency in mTOR or Rictor (Supplementary Fig. 1i–k). The trend for increased contraction was also observed in the aorta of $Rptor^{EC-/-}$ mice, though the difference was not statistically significant.

Electron paramagnetic resonance (EPR) spectroscopy was applied to measure the levels of NO which, once specifically trapped by cPTIO, causes a decrease in the EPR signal of cPTIO. Consistent with impaired EDD, the serum of $Mtor^{EC-/-}$, $Rptor^{EC-/-}$, or $Rictor^{EC-/-}$ mice amplified the cPTIO spectra which corresponded to a NO decrease of 21.6%, 18.2%, and 38.1%, respectively (Supplementary Fig. 1l–n). A similar reduction in NO of 28.8%, 29.0%, and 45.8%, respectively, was observed with a colorimetric measurement (Fig. 1g–i).

Reduced NO bioavailability was previously demonstrated to cause vascular dysfunction and set the stage for the initiation of atherosclerosis[19]. Moreover, an atherogenic diet increased the production of ROS, which reacted with and reduced the bioavailability of NO. To investigate the role of mTOR signaling in this context, $Apoe^{-/-}Mtor^{EC-/-}$, $Apoe^{-/-}Rptor^{enod-/-}$ and $Apoe^{-/-}Rictor^{EC-/-}$ mice were generated and fed with an atherogenic, high-fat diet. Serum NO was also decreased in these mice, but to a greater extent over that elicited by single gene deletion. The additional EC deficiency in mTOR, Raptor, and Rictor was associated with a reduction in NO of 31.1%, 35.4%, and 37.0%, respectively in the $Apoe$-null mice (Supplementary Fig. 1o–q).

These data clearly demonstrated that mTOR plays a key role in the basal regulation of EC-dependent vasomotor function. Inhibition of endothelial mTORC1 and/or mTORC2 causes impaired EDD, which is associated with decreased NO bioavailability, and exacerbated in the pro-atherosclerotic settings where DES is often utilized as a therapy.

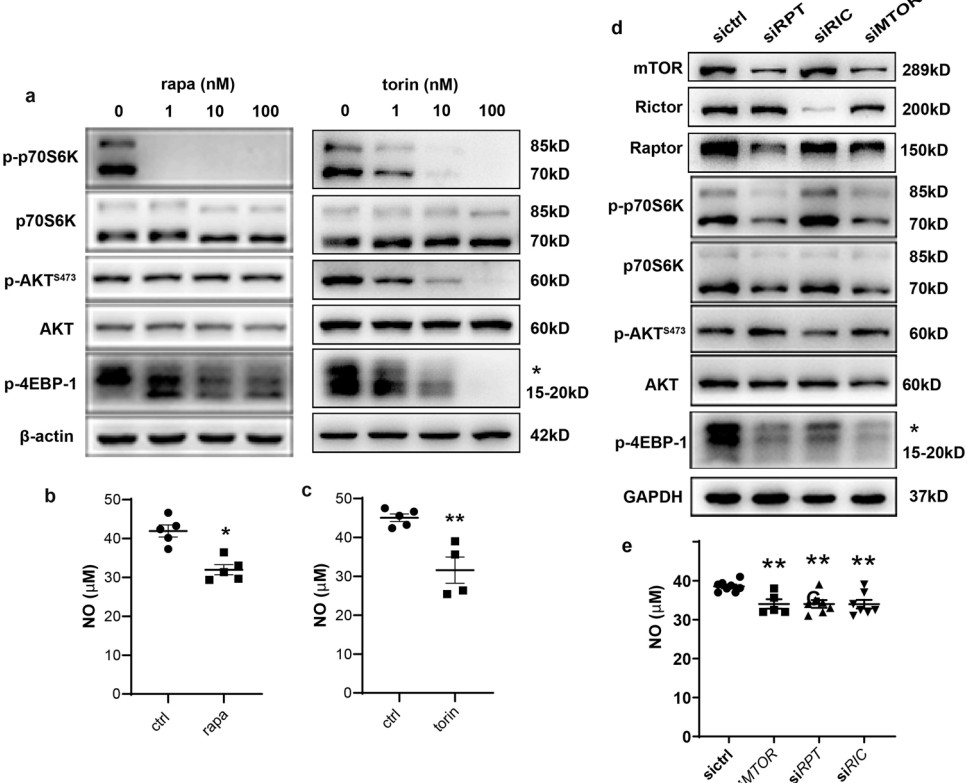

**Fig. 2 Both mTORC1 and mTORC2 contributed to NO bioavailability in HAEC. a** HAEC were treated with rapamycin (rapa) or torin 1 (torin) at indicated dose for 1 h prior to western blotting. *indicated the phosphorylated 4EBP-1. **b, c** NO levels of rapa (**b**) or torin-treated (**c**) HAEC were measured with colorimetric reaction assay. **d, e** At 48 h post transfection with control siRNA (sictrl), siRNA targeting *RPTOR (siRPT), RICTOR (siRIC)* or *MTOR (siMTOR)*, HAEC were submitted to western blot analysis (**d**) or NO measurement with colorimetric reaction assay (**e**) (*n* = 5–7). Shown images are representative blots of at least five experiments. **p* < 0.05; ***p* < 0.01 vs. ctrl or sictrl; unpaired two-tailed *t* test (**b, c**) or one-way ANOVA with Dunnett's post-test (**e**).

**Both mTORC1 and mTORC2 contributed to NO bioavailability in HAEC.** To investigate the roles of mTOR, mTORC1, and/or mTORC2 in signaling NO production by arterial EC, a pharmacological approach was utilized in which HAEC were exposed to rapamycin, an allosteric mTORC1 inhibitor, or torin 1, an inhibitor of both mTORC1 and mTORC2 for 1 h. Both inhibitors inactivated mTORC1 downstream phosphorylation of p70S6K and 4EBP-1, which was barely detectable in cells treated with 1 nM rapamycin or 10 nM torin 1 (Fig. 2a). Rapamycin at 1 nM, a concentration aligning closely with systemic concentrations measured in-stent–implanted patients[17], resulted in a reduction of NO production by ~25% as indicated by both EPR (Supplementary Fig. 2a) and colorimetric measurements (Fig. 2b). The similar effect on NO reduction was achieved by 10 nM torin 1 (Supplementary Fig. 2b, Fig. 2c). Therefore, clinically relevant concentrations of rapamycin (1 nM) and torin 1 (10 nM) was applied in the following studies.

Small interfering RNA (siRNA) was further employed to deplete *MTOR, RPTOR,* or *RICTOR* (Fig. 2d) in HAEC. Similar to the results in EC-specific gene knockout mice, inhibition of both complexes or a single disruption of either of the two complexes reduced NO release by ~15% in the culture medium (Fig. 2e). These results clearly demonstrated that both mTORC1 and mTORC2 contributed to NO bioavailability in HAEC.

**mTORC1 inhibition decreased eNOS expression.** To determine the source of mTOR-dependent increased NO bioavailability, the expression of eNOS, the main source of NO in EC, was examined. In response to rapamycin treatment, eNOS downregulation was observed (Fig. 3a), though the ratio of the phosphorylated form

(phosphorylation at Ser1177 or Thr495) to total eNOS which was associated with eNOS activity[11] was unaltered (Fig. 3b). Consistently, disruption of mTORC1 by silencing mTOR or Raptor expression decreased eNOS protein level in HAEC. However, the knockdown of Rictor to disrupt mTORC2 did not affect eNOS protein expression (Fig. 3c). Quantitative PCR indicated either pharmacologic inhibition (Fig. 3d) or siRNA-mediated disruption (Fig. 3e) of mTORC1 decreased *eNOS* transcription.

mTORC1 is known to affect translation mainly through phosphorylation of S6K1 and 4EBP-1. The activation of S6K1 leads to an increase in translational initiation and elongation as well as mRNA biogenesis. 4EBP-1 inhibits translation by binding to the translation initiation factor cap-binding protein eIF4E. The phosphorylation of 4EBP-1 disrupts this interaction, enabling the formation of the eIF4F complex which is required for the initiation of translation[4].

Here we utilized knockdown of the translation initiator *RPS6KB1* (encoding p70S6K) or overexpression of the translation repressor *EIF4BP-1* (encoding 4EBP-1) to specifically inhibit signaling downstream of mTORC1. Both caused a decrease in eNOS protein and mRNA (Fig. 3f, g). Together, these results demonstrated that signaling downstream of mTORC1 contributed to *eNOS* transcription and protein expression in HAEC.

**p70S6K downstream of mTORC1 contributed to *eNOS* transcription through posttranscriptional regulation of KLF2 expression.** Several transcription factors that putatively bind to the *eNOS* promoter were identified by analysis of the promoter in the JASPAR database, including Krüppel-like factor 2 (KLF2, LKLF), specificity protein-1 (Sp1), signal transducer, and activator of

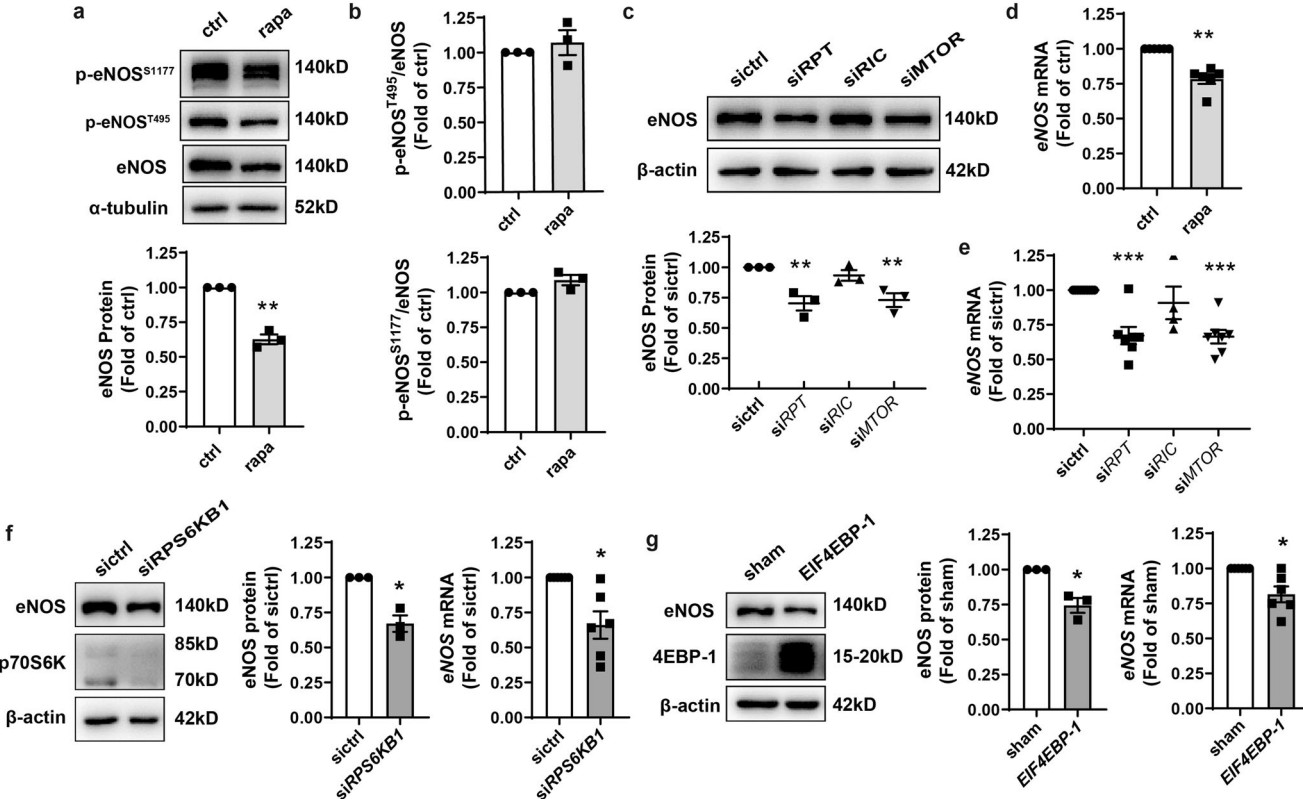

**Fig. 3 Inhibition of mTORC1 decreased eNOS expression. a, b** HAEC were treated with 1 nM rapa for 1 h prior to western blot analysis and quantification of eNOS protein ($n = 3$, **a**) as well as the ratio of phosphorylated Ser1177 or Thr495 to total eNOS (**b**). **c** HAEC were transfected with siRNA followed by western blotting and quantification of eNOS expression ($n = 3$). **d, e** HAEC were treated with 1 nM rapa for 1 h (**d**) or transfected with siRNA prior to quantitative PCR ($n = 5$–7). **f** At 48 h, post transfection with siRNA against *RPS6KB1*, HAEC were submitted to western blot analysis, quantification of eNOS expression ($n = 3$), and real-time PCR ($n = 6$). **g** HAEC were transfected with pcDNA3.1 (sham) or pcDNA3.1 *EIF4EBP-1* plasmids followed by western blot analysis, quantification of eNOS expression ($n = 3$) and quantitative PCR ($n = 6$). Error bars correspond to standard error of the mean (SEM). *$p < 0.05$; **$p < 0.01$; ***$p < 0.001$ vs. ctrl or sictrl or sham; one sample $t$ test (**a, b, d, f, g**) or repeated measures (RM) one-way ANOVA with Dunnett's test (**c, e**).

transcription 4 (STAT4), yin yang 1 (YY1), forkhead box protein P3 (FOXP3), activator protein 2α (AP2α) and interferon regulatory factor-1 (IRF-1). Except for IRF-1 (Supplementary Fig. 2c) and KLF2 (Fig. 4a), their expression remained unchanged with rapamycin treatment. IRF-1 was increased by mTOR inhibition, which was consistent with our previous observation[20]. KLF2 was reduced in rapamycin-treated (Fig. 4a) or mTORC1-disrupted (Fig. 4b) HAEC. This is consistent with previous reports that the ability of KLF2 to bind and activate the *eNOS* promoter is critical to the regulation of EC function[21]. To confirm its involvement in the regulation of *eNOS* transcription in HAEC, KLF2 expression was silenced with siRNA (Fig. 4c). Knockdown of KLF2 mRNA by 52.5% (Supplementary Fig. 2d) led to a 36.6% decrease in *eNOS* transcription (Fig. 4d) and a concomitant reduction in protein synthesis (Fig. 4c), which translated to a 35.8% reduction in NO release by HAEC (Fig. 4e). Surprisingly, *KLF2* transcription trended to increase rather than decrease with rapamycin treatment (Fig. 4f). Silencing of *RPS6KB1* enhanced while overexpression of *EIF4EBP-1* decreased *KLF2* transcription (Fig. 4g, Supplementary Fig. 2e) although both were associated with a downregulation on KLF2 protein (Fig. 4h, Supplementary Fig. 1I–f). These data suggested that at least p70S6K downstream of mTORC1 contributed to eNOS expression through posttranscriptional regulation of KLF2 expression.

**Nox-generated ROS-induced eNOS uncoupling in mTORC2-inhibited HAEC.** Reduced NO bioavailability and impaired EC-dependent vasorelaxation may also be caused by an excess of cellular ROS which converts NO-producing eNOS to a ROS generator[12]. To explain the normal eNOS expression (Fig. 3c) and decreased NO bioavailability (Fig. 2c) associated with mTORC2 inhibition, superoxide generation inside HAEC or release into mouse serum was assessed via DHE staining or by the chemiluminescent probe lucigenin, respectively. 1 h treatment with torin 1 rather than rapamycin increased superoxide accumulation by 23.6% in HAEC (Fig. 5a). Since prolonged treatment (24 h) with 100 nM rapamycin caused the disassembly of mTORC2 in HUVEC cells[7], we treated HAEC for 48 h and found that under these conditions rapamycin also increased superoxide (Supplementary Fig. 3a). ROS was also increased in the blood of mice treated with rapamycin as above (Supplementary Fig. 3b). Consistently, depletion of *RICTOR* or *MTOR* but not *RPTOR* increased superoxide generation by HAEC (Fig. 5b) or mouse primary EC (Fig. 5c), or release into the blood (Fig. 5d), indicating that inhibition or disruption of mTORC2 resulted in increased ROS generation. Moreover, measurement of the BH4 level in mTOR-inhibited HAEC indicated that torin 1 but not rapamycin treatment decreased BH4 (Fig. 5e). Finally, native PAGE followed by western blot revealed that the eNOS dimer to monomer ratio in torin 1-treated HAEC was decreased (Fig. 5f), a direct indicator of eNOS uncoupling[12,13]. Pretreatment with NOS inhibitor L-NAME (Supplementary Fig. 3c), attenuated torin 1-caused ROS overproduction by 14.1% (Fig. 5g). L-NAME is also an inhibitor of inducible NOS (iNOS). However, neither rapamycin nor torin 1 treatment affected iNOS expression

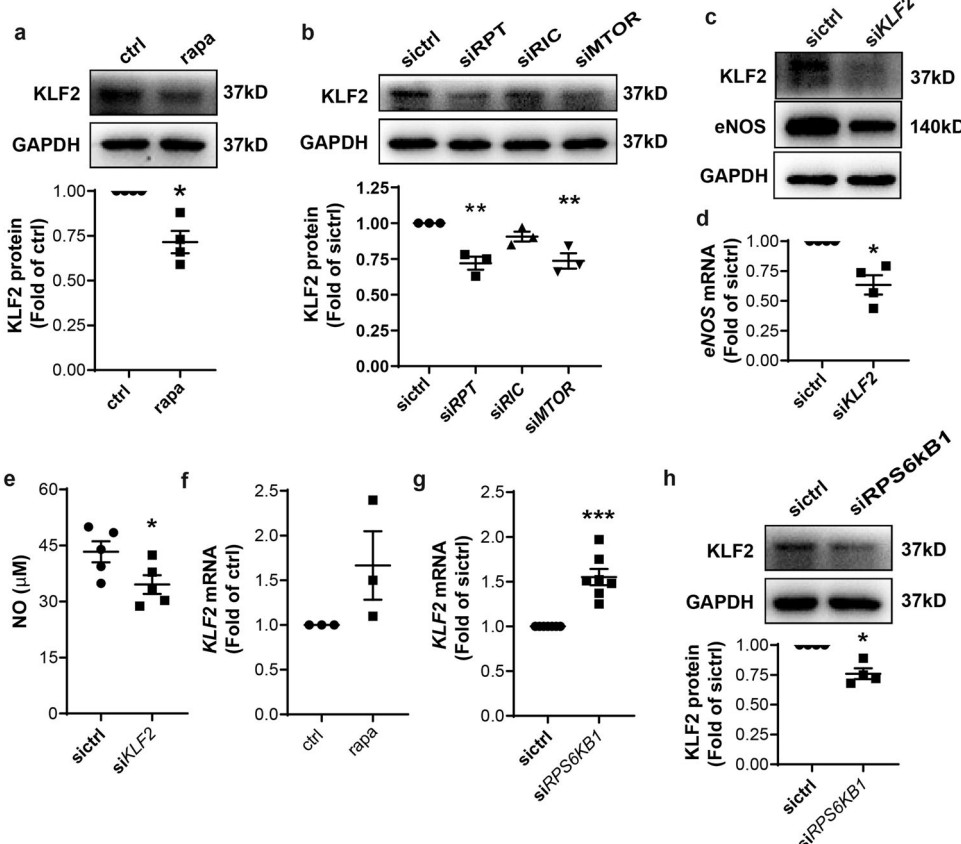

**Fig. 4 p70S6K downstream of mTORC1 contributed to *eNOS* transcription through translational regulation of KLF2 expression. a**, **b** HAEC were treated with 1 nM rapa or transfected with siRNA prior to Western blot analysis and quantification of KLF2 expression ($n = 3$–4). **c**, **d** HAEC were transfected with *KLF2*-targeting siRNA followed by Western blotting (c) and quantitative PCR ($n = 4$, **d**). **e** At 48 h post-transfection with siRNA against *KLF2*, relative NO level of the culture medium was measured with the colorimetric assay ($n = 5$). **f** HAEC were treated with 1 nM rapa for 1 h, prior to quantitative PCR to measure mRNA ($n = 3$). **g**, **h** HAEC were transfected with siRNA against *RPS6KB1* followed by quantitative PCR ($n = 7$, g) or western blotting ($n = 4$, **h**) to quantify KLF2 expression. $**p < 0.01$; $***p < 0.001$ vs. sictrl or ctrl; one-sample $t$ test (**a**, **d**, **f**, **g**, **h**) or RM one-way ANOVA with Dunnett's test (**b**) or unpaired two-tailed $t$ test (**e**).

(Supplementary Fig. 3d), excluding the involvement of iNOS in this process. Together these results confirmed the existence of eNOS uncoupling in mTORC2-inhibited cells.

In addition to uncoupled eNOS, the nicotinamide-adenine-dinucleotide phosphate (NADPH) oxidase (Nox), mitochondria, and xanthine oxidase (XO) constitute the major sources of cellular ROS[13]. The enhanced ROS release and impaired NO production caused by torin 1 treatment was reversed by nonspecific ROS scavenger N-acetylcysteine or the Nox inhibitor Apocynin, but not by the mitochondrion-targeting antioxidant MitoQ or XO inhibitor Allopurinol (Fig. 5h, i, Supplementary Fig. 3e).

Nox is a key ROS-producing enzyme, which transfers electrons from NADPH to molecular oxygen, thereby generating super-oxide. Nox-derived ROS causes oxidation of BH$_4$, resulting in eNOS uncoupling[22]. The family of Nox contains seven members, of which Nox2 has emerged as the most widely expressed Nox isoform in vascular EC[23]. The upregulation of its protein directly accounted for the increased Nox-dependent superoxide produc-tion. Torin 1 induced both the transcription and protein expression of Nox2 (previously known as gp91$^{phox}$) which is the specific subunit of the identically named enzyme Nox2 (Fig. 5j–k). Increased Nox2 protein was also observed in *RICTIR* or *MTOR*-silenced HAEC (Fig. 5l). Taken together, these results implicate Nox2 as predominantly responsible for the increase in ROS and concomitant eNOS uncoupling elicited by mTORC2 inhibition.

**Crosstalk between MAPK (p38 and JNK) activation and Nox2 upregulation contributed to eNOS uncoupling in response to mTORC2 inhibition.** PKC-dependent activation of Nox might be essential to the mechanism responsible for increased oxidative stress in diabetic vascular tissues[24]. Our previous study reported that chemical inhibitors or siRNA-mediated mTOR knockdown activated PKC[25]. Therefore, we investigated the role of PKC in linking mTOR inhibition with ROS production. However, torin 1-induced ROS production in HAEC was not decreased by treatment with either Gö−6976 (inhibitor of conventional PKCs) or Rottlerin (inhibitor of PKCδ) (Supplementary Fig. 3f).

mTOR inhibition was previously associated with the activation of mitogen-activated protein kinase (MAPK) cascades[26] which were involved in ROS generation[27]. At the doses applied in this study treatment of HAEC with torin 1 but not rapamycin increased phosphorylation of MAPK p38 (p38), c-Jun N-terminal kinase (JNK), and extracellular signal-regulated kinases 1 and 2 (ERK1/2) (Fig. 6a). Similarly, enhanced activation of p38 and JNK in HAEC was associated with knockdown of *MTOR* or *RICT*OR to disrupt mTORC2 compared to knockdown of *RPTOR* to inhibit mTORC1, although the phosphorylation of p38 and JNK was also slightly increased in the latter (Supplementary Fig. 3g).

Pharmacological inhibition of p38 or JNK activity using SB 203580 or SP 600125, but not ERK1/2 using U0126, decreased torin 1-induced superoxide production (Fig. 6b), accompanied by a reversal in NO reduction (Fig. 6c). It is noteworthy that

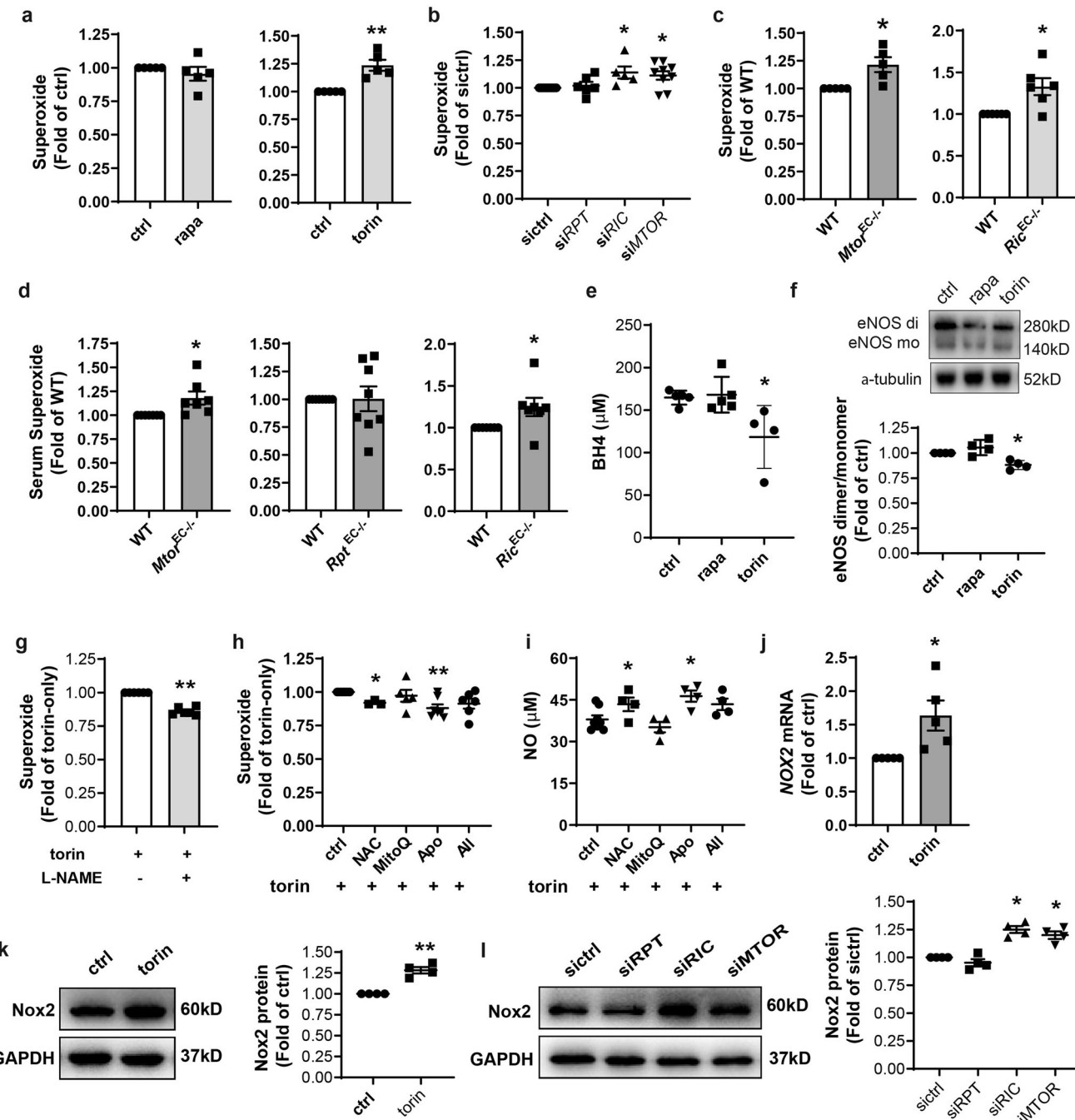

**Fig. 5 Inhibition of mTORC2 promoted ROS accumulation and eNOS uncoupling. a** HAEC were treated with 1 nM rapa or 10 nM torin for 1 h followed by incubation with dihydroethidium (DHE) for detection of intracellular superoxide with flow cytometry ($n = 5$). **b** HAEC were transfected with siRNA prior to flow cytometric measurement of intracellular superoxide ($n = 5$-9). **c** Primary endothelial cells (EC) were isolated from $Mtor^{EC-/-}$ or $Rictor^{EC-/-}$ mice and their WT littermates to measure intracellular superoxide ($n = 5$-6). **d** Serum superoxide levels were measured in $Mtor^{EC-/-}$, $Rpt^{EC-/-}$ and $Ric^{EC-/-}$ mice with chemiluminescent assay ($n = 7$-9). **e** HAEC were treated with 1 nM rapa or 10 nM torin for 1 h followed by ELISA measurement of BH4 ($n = 4$-5). **f** After treatment, eNOS dimer (di) to monomer (mo) ratio was evaluated by native PAGE followed by western blotting ($n = 4$). **g** After pretreatment with L-NAME (500 nM) for 0.5 h ($n = 3$-6), HAEC were incubated with 10 nM torin for 1 h prior to measurement of intracellular superoxide with flow cytometry. **h, i** After 1 h pretreatment with N-acetylcysteine (Nac, 2 mM), MitoQ (1 µM), Apocynin (Apo, 20 µM), or Allopurinol (All, 50 µM) ($n = 3$-4), HAEC were incubated with 10 nM torin for 1 h prior to measurement of superoxide inside HAEC (**f**) and NO in the medium (**g**). **j** HAEC were treated with 10 nM torin for 1 h prior to quantitative PCR ($n = 5$). **k, l** HAEC were treated with 10 nM torin or transfected with siRNA followed by Western blotting to quantify Nox2 expression ($n = 4$). Error bars correspond to standard error of the mean (SEM). *$p < 0.05$; **$p < 0.01$ vs. ctrl, WT, sictrl or torin-treated ctrl; one-sample $t$ test (**a**, **c**, **d**, **g**, **j**, k) or RM one-way ANOVA with Dunnett's test (**b**, **f**, **h**, **l**) or one-way ANOVA with Dunnett's test (**e**, **i**).

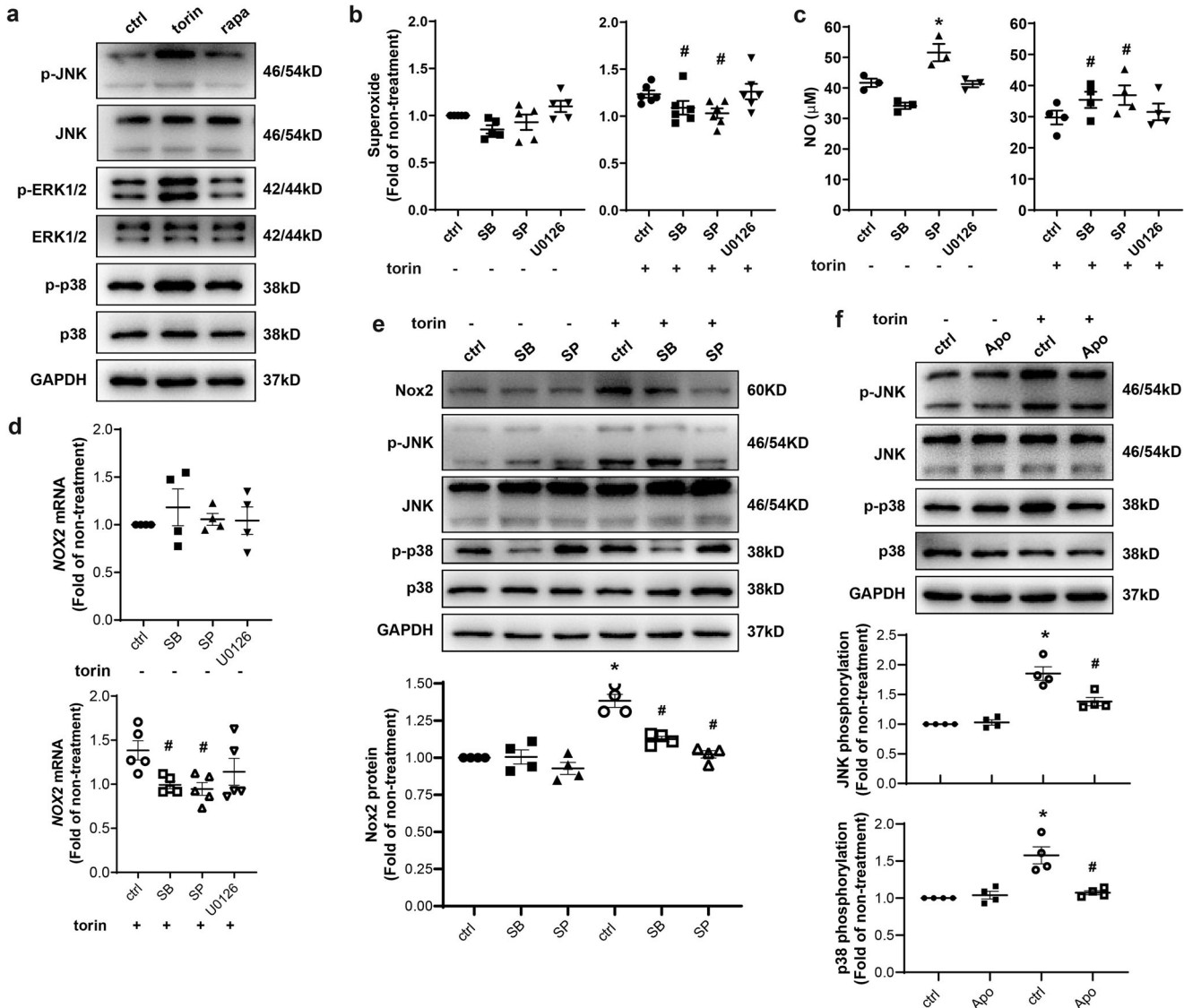

**Fig. 6 Crosstalk between MAPK (p38 and JNK) activation and Nox2 upregulation contributed to excessive ROS production in mTORC2-inhibited HAEC. a** After treatment with 10 nM torin or 1 nM rapa for 1 h, HAEC were submitted to western blot analysis. **b–d** HAEC were pretreated with SB 203580 (SB, 20 μM), SP 600125 (SP, 20 μM) or U0126 (5 μM) for 1 h before incubation with or without 10 nM torin for 1 h prior to DHE staining ($n = 5$, b), NO assay ($n = 3–4$, **c**) or quantitative PCR ($n = 4–5$, **d**). **e** HAEC were treated with SB (20 μM) or SP (20 μM) for 1 h before incubation with 10 nM torin for 1 h prior to Western blotting and quantification of Nox2 expression. **f** After pretreatment with Apo (20 μM) for 1 h, HAEC were incubated with or without 10 nM torin for 1 h before western blot analysis and quantification of JNK and p38 phosphorylation. *$p < 0.05$ vs. untreated ctrl; #$p < 0.05$ vs. torin-treated ctrl; RM one-way ANOVA with Dunnett's test (**b–d**) or Tukey's test (**e**, **f**).

inhibition of JNK also increased basal NO release. Consistent with these results, blocking either p38 or JNK resulted in a reduction in torin 1-induced Nox2 expression both at mRNA (Fig. 6d) and protein (Fig. 6e) levels. Interestingly, inhibition of Nox by Apocynin, in turn abolished the torin 1-induced activation of p38 and JNK (Fig. 6f).

These results implicate a positive feedback between MAPK (p38 and JNK) activation and Nox2 upregulation that contributes to ROS generation and eNOS uncoupling caused by mTORC2 inhibition.

**Modulation of KLF2 or Nox2 expression restored EDD function in endothelial mTORC1- or mTORC2-inhibited mice.** In accordance with the findings in HAEC, expression of KLF2 and eNOS were downregulated in primary EC isolated from $Rptor^{EC-/-}$ mice

(Fig. 7a), whereas Nox2 expression increased in EC from $Rictor^{EC-/-}$ mice (Fig. 7b). These results support the possibility that targeting these pathways could improve EDD under mTOR inhibition in vivo. To demonstrate the feasibility of such an approach murine EC were infected with AAV which contained either $Klf2$-coding or $Nox2$-targeting sequence fused with $EGFP$ and $Flag$ whose expression was driven by the EC-specific $ICAM2$ promoter. Western blot analysis confirmed successful overexpression of KLF2 and the subsequent eNOS increase in murine EC infected with $Klf2$-expressing AAV. Moreover, overexpression prevented the decrease in KLF2-induced eNOS caused by rapamycin (Fig. 7c). In contrast, infection with $Nox2$-targeting AAV downregulated Nox2 expression (Fig. 7d).

After i.v. injection of $Klf2$-expressing AAV to $Rptor^{EC-/-}$ mice, or $Nox2$-targeting AAV to $Rictor^{EC-/-}$ mice, EC-specific expression was confirmed by fluorescent microscopy observation of EGFP co-localization with CD31 (Supplementary Fig. 4a).

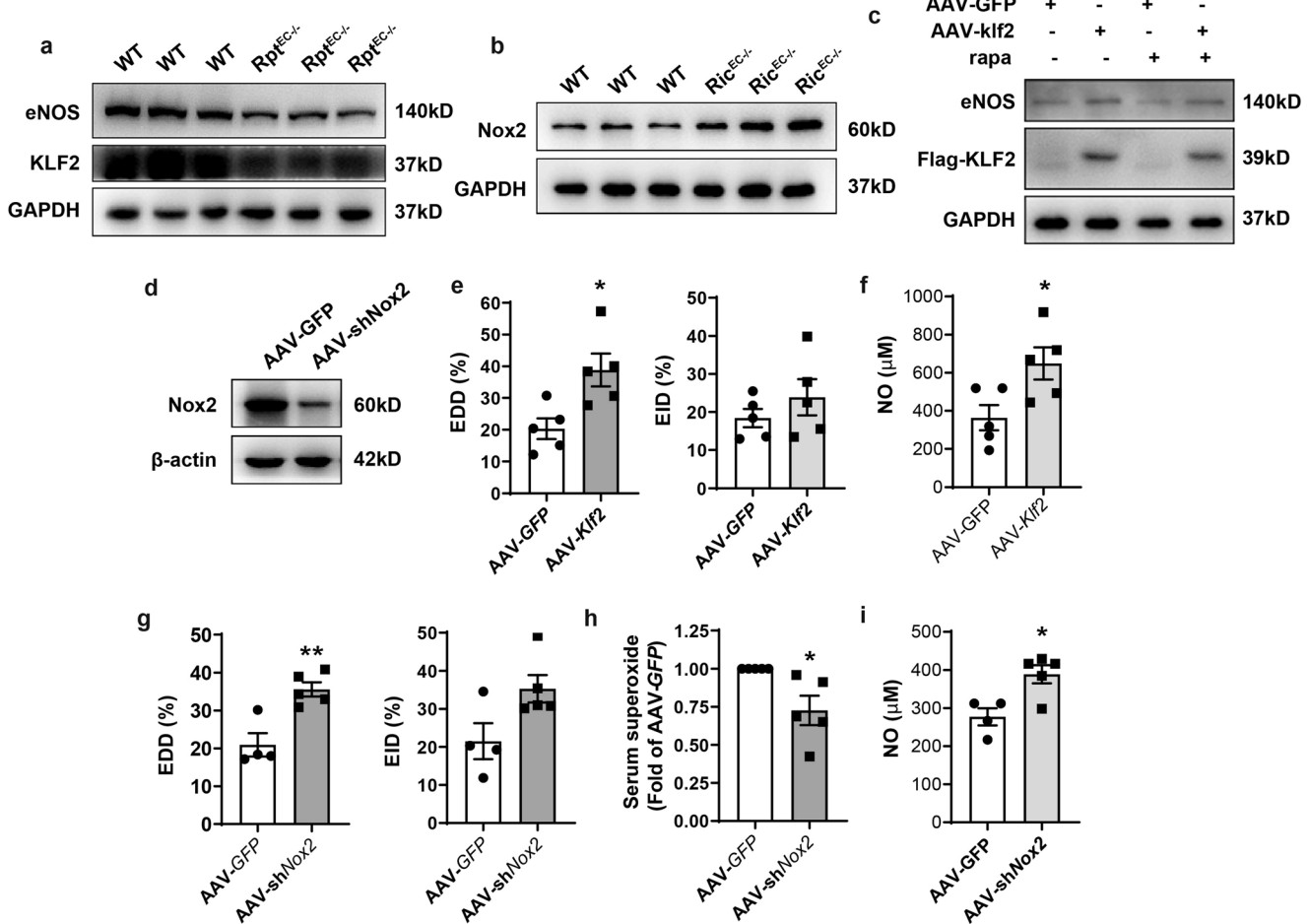

**Fig. 7 Modulation of KLF2 or Nox2 expression restored EDD function in endothelial mTORC1- or mTORC2-inhibited mice. a, b** EC were isolated from $Rptor^{EC-/-}$ or $Rictor^{EC-/-}$ and their WT littermates followed by western blotting. **c** EC isolated from wild-type mice were infected with empty AAV (AAV-GFP) or $Klf2$-expressing adeno-associated virus (AAV-$Klf2$) at a multiplicity of infection (MOI) of $10^5$ vg/cell for 72 h, then treated with or without 1 nM rapa for 1 h followed by western blot analysis. **d** Murine EC was infected with AAV containing two tandem mouse Nox2-targeting sequences (AAV-sh$Nox2$) followed by western blot analysis. **e, f** $Rptor^{EC-/-}$ mice were i.v. injected with AAV-GFP or AAV-$Klf2$ at $3 \times 10^{11}$ vg. EDD and EID were evaluated with transcutaneous ultrasound imaging ($n = 5$, **e**), and serum NO evaluated by the colorimetric measurement ($n = 5$, **f**). **g–i** $Rictor^{EC-/-}$ mice were injected with $3 \times 10^{11}$ vg AAV-GFP or AAV-sh$Nox2$ followed by evaluation of EDD and EID ($n = 5$, **g**), and evaluation of serum superoxide ($n = 5$, **h**) and NO by the colorimetric measurement ($n = 4$, **i**). Error bars correspond to the standard error of the mean (SEM). *$p < 0.05$; **$p < 0.01$ vs. AAV-GFP; unpaired two-tailed $t$ test (**e–g**, **i**) or one sample $t$ test (**h**).

Injection of AAV modulated eNOS and Nox2 expression specifically in the intima (Supplementary Fig. 4b, c), validating the EC specificity driven by ICAM2 promoter.

Ultrasound imaging showed that EC-targeting overexpression of KLF2 restored FMD but not EID in $Rptor^{EC-/-}$ mice (Fig. 7e, Supplementary Fig. 4d), concomitant with an increase in the serum NO level (Fig. 7f, Supplementary Fig. 4e). Likewise, silencing $Nox2$ remarkably offset FMD decrease in Rictor $^{EC-/-}$ mice (Fig. 7g, Supplementary Fig. 4f). It also trended to increase EID (Fig. 7g) although no significance was achieved. Consistently, the enhanced superoxide release and impaired NO production in the serum of $Rictor^{EC-/-}$ mice were reversed by infection of $Nox2$-targeting AAV (Figs. 7h, i, Supplementary Fig. 4g).

## Discussion
Although mTOR inhibitors have been used effectively in combination with stenting in coronary interventions due to their anti-proliferative properties[1], a deleterious effect of rapamycin and its analogs on vessel healing and EC-dependent vasoreactivity has been reported[2,3,5]. Consistently, deficiency of SMC[28] or

endothelial[29] mTORC1 resulted in impaired vascular relaxation. Herein we examined the mechanism linking inhibition of endothelial mTORC1 or mTORC2 with altered NO metabolism leading to this impaired EC function. Using a combination of endothelial-specific deletions targeting the mTOR signaling pathways in mice and complementary studies in cultured endothelium, we report that mTOR inhibition acts through different but synergistic mechanisms to affect NO bioavailability and EC-dependent vasoreactivity. Specifically, inhibition of mTORC1 signaling resulted in decreased KLF2-dependent eNOS expression, whereas inhibition of mTORC2 resulted in Nox2-dependent ROS accumulation and eNOS uncoupling (Fig. 8).

In contrast to recent studies mainly using the aortic or mesenteric rings[29,30], in this study we applied in vivo non-invasive transcutaneous ultrasound imaging of the abdominal aorta and ex vivo analysis of aortic rings of $Mtor^{EC-/-}$, $Rptor^{EC-/-}$, and $Rictor^{EC-/-}$ mice to assess the contribution of EC mTOR signaling to vasorelaxation. FMD rather than GTN-induced vasodilatation of the abdominal aorta was affected, demonstrating that the impaired relaxation was EC-dependent. Consistent with the in vivo findings, the aortic rings of $Mtor^{EC-/-}$, $Rptor^{EC-/-}$ and $Rictor^{EC-/-}$ mice

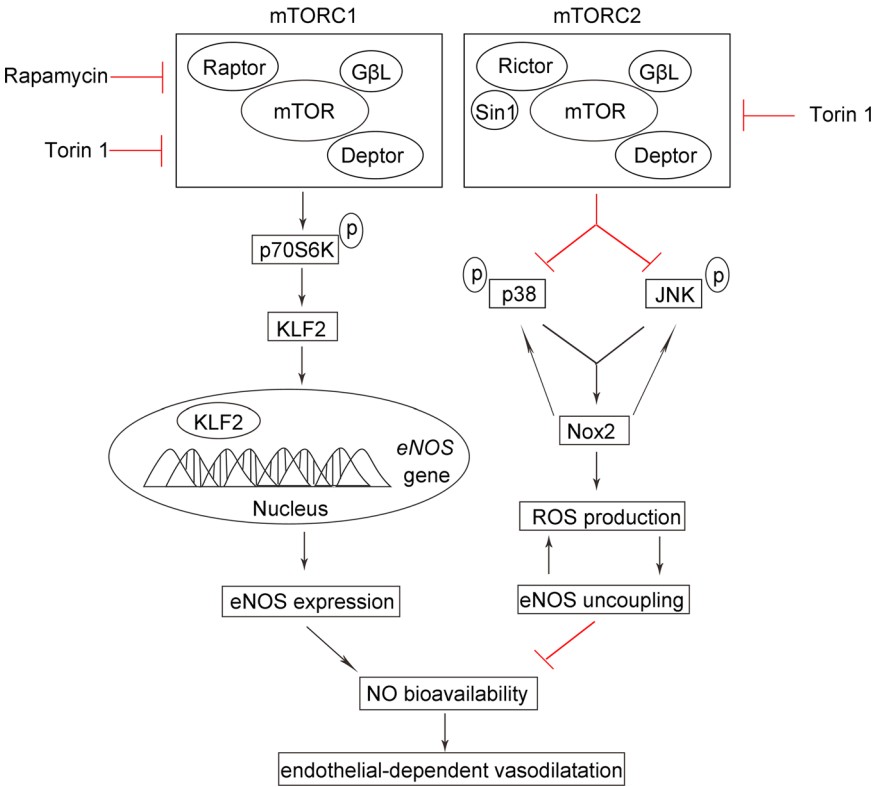

**Fig. 8 Schematic illustrating the regulation of NO bioavailability and EDD function by mTOR signaling.** Inhibition of mTORC1 suppresses *eNOS* gene expression, by impairing p70S6K-mediated posttranscriptional regulation of the transcription factor KLF2 expression. In contrast to *mTORC1* inhibition, targeting mTORC2 does not affect eNOS expression. Rather, disruption of mTORC2 results in activation of MAPK (p38 and JNK), which are normally suppressed. A positive-feedback loop with Nox2 upregulation is enabled that contributes to the excessive generation of reactive oxygen species (ROS), leading to eNOS uncoupling and decreased NO bioavailability in mTORC2-inhibited EC.

displayed diminished vasodilatory responses to ACh. Moreover, the deficiency of mTOR or Rictor in EC intensified the aortic contraction in response to NE, indicating that mTOR-mediated NO production by EC contributes basally to the regulation of vessel tone. This is consistent with the revelation that the NO pathway at the EC level has the capacity to counter-balance the effect of NE[31].

One of the most important EDRF, NO plays a central role in the maintenance of vasoreactivity and the prevention of atherogenesis[19]. The NO level was decreased in the serum of all $Mtor^{EC-/-}$, $Rptor^{EC-/-}$ and $Rictor^{EC-/-}$ mice. Deficiency of mTOR, Raptor, or Rictor exacerbated the NO reduction in the *Apoe*-null mice, illustrating the putative importance of impaired mTOR signaling in the presence of dyslipidemia that promotes atherosclerosis. Inhibition of mTOR or disruption of either complex in EC may further promote the development of atherosclerosis, which is the subject of our ongoing study.

In line with our findings of the involvement of both mTOR complexes in NO-dependent vasorelaxation, blood pressure was increased in mice with EC-specific *Rptor* deletion both before and after angiotensin II (ANG II) infusion[28], and deletion of *Rictor* and thereby inactivation of mTORC2 in perivascular adipose tissue increased aortic contraction and impaired vasorelaxation[14].

eNOS, the predominant NOS isoform expressed constitutively in EC, is responsible for most of the NO production. In a study of murine carotid arteries, rapamycin was reported to reduce eNOS expression during high shear stress conditions, a situation that may be relevant to the up- and downstream flow disturbances in the region of the DES[32]. However, other groups reported no change in eNOS expression in the aortic lysates prepared from EC Raptor-deficient mice[29,30], or even an increase in eNOS expression in the aorta of rapamycin-treated mice[33]. Our study with

both primary mouse EC culture and primary HAEC revealed that inhibition of mTORC1 effectively suppressed *eNOS* transcription through downregulation of KLF2.

KLF2 is a member of the zinc finger family of transcription factors, which mediate anti-inflammatory, anti-thrombotic, and anti-proliferative responses in EC. Consistent with previous reports that KLF2 transactivated *eNOS* and contributed to EC-dependent vasorelaxation[21,34], silencing of KLF2 decreased eNOS expression as well as secreted NO in this study. We revealed that p70S6K promoted the eNOS/NO pathway through post-transcriptional regulation of KLF2 expression, which was in line with the primary function of mTORC1 downstream signaling in protein synthesis. Surprisingly, we observed increased *KLF2* transcription associated with p70S6K inhibition. In this regard, active mTORC1 was reported to directly phosphorylate Foxo, a transcription factor of KLF2, on multiple residues. After phosphorylation, Foxo was maintained within the cytoplasm, and thus the transcription of *KLF2* repressed[35]. The increase in *KLF2* transcription may therefore represent a compensatory mechanism of EC to maintain homeostasis.

Although rapamycin post-transcriptionally downregulated KLF2 expression, infection with *Klf2*-expressing AAV was sufficient to restore its expression in murine EC and subsequently rescue EC-dependent vasodilation in $Rptor^{EC-/-}$ mice, strengthening the causal relationship between decreases in KLF2-induced eNOS and impairment of EDD in the mTORC1-inhibited artery.

In addition to regulation of expression, eNOS activity was subject to post-translational modification by mTOR inhibition, although the results were conflicting. AKT and PKC that could be activated by mTORC2 signaling[4] were reported to phosphorylate on Ser1177 and Thr495 of eNOS, respectively[11]. Silencing

*RICTOR* indeed inhibited AKT activity indicated by Ser473 phosphorylation; however, supporting the observations in muscle cells[36] and cancer cells[37], we found in HAEC that AKT activity was never inhibited by the interference of mTOR to disrupt mTORC1/2. The discrepancies in AKT phosphorylation by depletion of mTOR could be due to variable cell types applied in different studies. Nonetheless, our results indicated that AKT signaling was unlikely a shared mechanism by mTORC2 inhibition to affect eNOS activity in *RICTOR*- and *MTOR*-depleted HAEC. Infusion of mice with rapamycin-induced an intracellular $Ca^{2+}$ leak and increased conventional PKC-mediated eNOS phosphorylation at Thr495 which was associated with low eNOS activity and decreased NO production[15]. On the contrary, rapamycin was also observed to increase eNOS phosphorylation at Ser1176 (corresponding to Ser1177 of human eNOS) and hence its activity in the murine cortical vessels[38]. Our results, however, indicated that phosphorylation at either Ser1177 or Thr495 was proportional to the total eNOS in either rapamycin-treated HAEC, excluding the regulation of NO production by phosphoryl-modification of eNOS.

Under oxidative stress which is commonly associated with ROS overproduction and inflammation, the NO-producing eNOS is converted to $O_2^-$-generating enzyme through the process of eNOS uncoupling. Pro-inflammatory cytokines (e.g. TNF-α and IL-1β) known to activate Nox are associated with elevated ROS production and oxidative stress. Even worse, increased eNOS aggravates this condition. Consistently, our in vivo study found that the deficiency in mTOR, Raptor, or Rictor exacerbated the NO reduction in atherosclerotic mice characterized by systemic inflammation. Our laboratory has also found an increased transcription of pro-inflammatory cytokine genes such as *Tnfα*, *Il1α*, *Il1β* in *Rictor*$^{EC-/-}$ and *Mtor*$^{EC-/-}$ mice (submitted data), confirming a pro-inflammatory role for mTORC2 inactivation.

Studies have implicated mTOR signaling in the regulation of ROS generation. Activation of mTORC1 in EC was associated with ROS generation[39]. As mTORC1 activity is well known to negatively regulate autophagy[4], this result was consistent with the observation that inhibition of autophagy by knocking down Atg3[40] or Atg7[41] aggravated oxidative stress and decreased NO production in EC. However, in this study, we found in HAEC no change in ROS generation by mTORC1 inhibition either in acute rapamycin treatment or in *RPTOR* depletion, implying that activation of autophagy might be insufficient to suppress oxidative stress. In line with the result that inactivation of mTORC2 by depletion of *Rictor* in adipose tissue promoted inflammation and oxidative stress[14], we found inhibition of mTORC2 promoted p38/JNK-mediated Nox2 expression causing ROS overgeneration and impaired NO bioavailability. More importantly, ROS was increased in *Mtor*$^{EC-/-}$ and *Rictor*$^{EC-/-}$ mouse serum, implying that ROS could serve as a marker of mTORC2 inactivation. Furthermore, we revealed here crosstalk between MAPK (p38 and JNK) activation and Nox2 upregulation. While our results with torin 1 were consistent with previous reports documenting an association of mTOR inhibition with ROS-promoting MAPK activation[26,27], we failed to observe p38 and JNK activation in rapamycin treatment. One possible reason might be that the clinically relevant low dose of rapamycin used in this study was insufficient to elicit such activation. Nevertheless, our data indicated that in HAEC p38 and JNK activation might be more sensitive to inhibition of mTORC2.

Among the four enzyme systems producing ROS, Nox appeared to be the predominant source of ROS in the vasculature[13]. The non-selective Nox inhibitor Apocynin elicited antihypertension and vasodilator effects in different models[42–44] through inducing both EC- and SMC-dependent mechanisms. As the most widely expressed Nox isoform in EC[23], Nox2 plays a major role in the regulation of EC functions, by affecting NO bioavailability and modulating the expression of adhesion molecules during inflammation and angiogenesis[23]. As revealed by a cross-sectional study, the blood level of its soluble form was inversely correlated with FMD[45]. EC-targeted overexpression of Nox2 attenuated ACh-induced vasorelaxation[46]. Consistently, our study revealed that EC-specific knockdown of *Nox2* restored FMD in *Rictor*$^{EC-/-}$ mice.

Chronic release of the eluted drug in the stent may also disrupt the mTORC2 complex[7], raising the possibility that both eNOS downregulation and uncoupled eNOS contributes to late in-stent vasodilatory dysfunction. This is supported by our results that increased ROS was observed in both HAEC and mouse blood after prolonged treatment with rapamycin. Moreover, rapamycin, as well as defective NO bioavailability[47], contribute to impaired EC recovery which may also factor into EC-dependent vascular dysfunction associated with implantation of DES. In this regard, repeated studies of vasomotor function after DES implantation at later time points to evaluate the time frame of EC recovery will ultimately be needed.

In conclusion, this study revealed that mTOR inhibition reduced eNOS expression and activity, which was dependent on mTORC1- and mTORC2-mediated signaling, respectively. These pathways synergistically caused reduced NO bioavailability and impaired endothelium-dependent vasorelaxation in mTOR-inhibited HAEC and mice. Moreover, animal studies confirmed the feasibility of targeting the underlying pathways for possible therapeutic benefit in rescuing impaired endothelial function under mTOR inhibition. Further studies may facilitate the development of novel therapeutic strategies to reduce the side effects of DES widely used in coronary interventions.

## Methods

**Chemicals and reagents**. Antibodies targeting mTOR (#2983), Rictor (#2114), Raptor (#2280), p-eNOS (S1177) (#9570), IRF-1 (#8478), AP2α (#3215), p-p70S6K (Ser389) (#9234), p70S6K (#9202), p-4EBP-1 (T37/46) (#2855), 4EBP-1 (#9452), p-AKT (S473) (#4060), AKT (#4691), p-p38 (Thr180/Tyr182) (#4511), p38 (#8690), p-ERK (Thr202/Tyr204) (#4370), ERK (#4695), p-SAPK/JNK (Thr183/Tyr185) (#4668), JNK (#9252), GAPDH(#2118), α-Tubulin (#3873) and eNOS (#32027) were from Cell Signaling Technology. Antibodies targeting eNOS (sc-376751), p-eNOS (T495) (sc-136519), gp91phox (sc-130543), Sp1 (sc-17824), p-STAT4 (S721) (sc-28296), YY1 (sc-7341), FOXP3 (sc-166212), Nox2 (sc-130543) were purchased from Santa Cruz Biotechnology. Anti-KLF2 was from Abcam (ab139699) or Bioss (bs-2772r). Antibody targeting iNOS was from Abcam (ab178945). Anti-β-actin was purchased from Bioss (bs-0061r-100). Anti- CD31 was from BD Pharmingen (553370). Alexa 594-conjugated goat anti-rat IgG was from Invitrogen (A11007) and PE-conjugated mouse anti-rabbit IgG from Santa Cruz Biotechnology (sc-3753). NOS inhibitor L-NAME (N5751, Sigma-Aldrich), PKCδ inhibitor Rottlerin (R5648) and PKCα and PKCβI inhibitor Gö−6976 (A43) were from Sigma-Aldrich. ROS scavenger N-acetylcysteine (HY-B0215), mitochondrial antioxidant MitoQ (HY-100116A), Nox inhibitor Apocynin (HY-N0088), XO inhibitor Allopurinol (HY-B0219), p38 inhibitor SB 203580 (HY-10256) and JNK inhibitor SP 600125 (HY-12041) were purchased from MCE. ERK inhibitor U0126 was from Selleck (U120).

**Cell culture and treatment**. Primary HAEC were purchased from American Type Culture Collection (Cat No. PCS-100-011, Lot No. 63233442, Manassas, VA) and maintained in endothelial growth medium-2 (Lonza) supplemented with 10% fetal bovine serum (FBS) (Gibco) at 37 °C and 5% CO₂. HAEC have treated with mTOR inhibitors rapamycin (V900930, Sigma-Aldrich) or torin 1 (14379, Cell Signaling Technology) before NO or ROS measurement, western blot analysis, or quantitative PCR. When necessary, HAEC were pretreated with pharmacological inhibitors.

**Cell transfection**. Negative control siRNA (sc-37007), siRNA targeting *KLF2* (sc-35818), *RPTOR* (encoding Raptor) (sc-44069), *RICTOR* (sc-61478), *MTOR* (sc-35409), *RPS6KB1* (encoding p70S6K) (sc-36165) were purchased from Santa Cruz Biotechnology (Santa Cruz, CA). pcDNA3.1 *EIF4EBP1* (encoding 4EBP-1) was constructed using PCR with primers containing restriction enzyme sites. siRNA or plasmid was transfected into HAEC using Lipofectamine 2000 (Thermo Fisher Scientific, USA) according to the manufacturers' instructions.

**Western blot analysis**. Cells were collected in lysis buffer. After boiling for 5 min, the total lysate was separated by SDS-PAGE and transferred onto PVDF membranes (Millipore). For native PAGE, SDS was not contained in the lysis buffer, gel composition, or electrophoresis. Nonspecific binding of antibody was blocked with 1% BSA and then incubated with the 1:1000 diluted primary antibody. After incubated with the appropriate secondary antibody, the protein bands were visualized with ECL (FD8020, FDBIO) and detected using a digital gel image analysis system (Tanon, Shanghai, China). Shown in this study were representative images from at least three independent experiments.

**Quantitative PCR**. Total RNA extracted from HAEC with TRIzol reagent (Takara Biotechnology) was used to synthesize 1st strand cDNA with cDNA Synthesis Supermix for qPCR (Yeasen Biotech) according to the manufacturer's instructions. SYBR Green (Yeasen Biotech) was used for real-time PCR on a Light Cycler 480 Instrument II (Roche). The relative fold change was calculated by normalizing to $GAPDH$ using the $2^{-\Delta\Delta Ct}$ method. The primers for mRNA measurement were as follows: $eNOS$, forward: 5'-TTCCGCTACCAGCCAGACC-3' and reverse: 5'-CACTCGCTTCGCCATCACC-3'; $KLF2$, forward: 5'-CAAGACCTACACCAAGAGTTCG-3' and reverse: 5'- CATGTGCCGTTTCATGTGC-3'; $NOX2$, forward: 5'-CAGCTATGAGGTGGTGATG-3' and reverse: 5'-GCCAGTGAGGTAGATGTTG-3'; $GAPDH$, forward: 5'-GGATTTGGTCGTATTGGG-3' and reverse: 5'-GGAAGATGGTGATGGGATT-3'.

**BH4 assay**. The level of BH4 in HAEC treated with rapamycin or torin 1 for 1 h was detected by enzyme-linked immunosorbent assay (ELISA) kit according to the manufacturer's instructions (JL15170, Shanghai Jianglai Biotechnology, China).

**Rapamycin treatment of mice and establishment of EC-specific deficiency**. All animal procedures were approved by the institutional animal care and use committee (IACUC-1908030). 6-week-old WT mice were intraperitoneally injected with vehicle (corn oil) or 2 mg/ kg body weight/ day rapamycin (HY-10219, MCE) for 7 consecutive days, which led to a comparable blood concentration of rapamycin in mice to that in patients implanted with DES[17]. $Mtor^{flox/flox}$, $Rptor^{flox/flox}$ and $Rictor^{flox/flox}$ mice were crossed with $Cdh5Cre^{ERT2}$ mice[48] to generate EC-specific knockout mice (positive for $Cdh5Cre^{ERT2}$ allele, also referred as $Mtor^{EC-/-}$, $Rptor^{EC-/-}$ and $Rictor^{EC-/-}$) and their wild type littermate controls (negative for $Cdh5Cre^{ERT2}$ allele, referred as WT). Sex-matched EC-specific knockout mice and respective WT controls at 6-week old received intraperitoneal injections of tamoxifen (135 mg/kg, dissolved in corn oil) once per day for 7 consecutive days to induce EC-selective deletion of $Mtor$, $Rptor$ or $Rictor$ in $Mtor^{EC-/-}$, $Rptor^{EC-/-}$ or $Rictor^{EC-/-}$ mice. The mice were used two weeks after the last tamoxifen injection.

To evaluate the effects of mTOR inhibition in the pro-atherosclerotic settings, ApoE deficiency was further introduced to generate $Apoe^{-/-}Mtor^{EC-/-}$, $Apoe^{-/-}Rptor^{EC-/-}$ and $Apoe^{-/-}Rictor^{EC-/-}$ mice, and $Apoe$ single knockout littermate controls, of which the males at 6-week old were injected with tamoxifen as above and put on a high-fat diet containing 1.25% cholesterol (Research Diet, D12108C) for 12 weeks.

**Transcutaneous ultrasound measurement of vasomotor function**. The non-invasive transcutaneous ultrasound evaluation of endothelial function was performed as described previously[49] with modification. After anesthetized with an intraperitoneal injection of pentobarbital sodium (Sigma-Aldrich) at a dose of 60 mg/kg body weight, mice were placed in dorsal decubitus on a heater blanket and the abdomen, as well as the groin, was shaved, and lubricated with ultrasound gel. The heart rate and diameter of abdominal aorta were measured with B-mode ultrasound images (Vevo 770, Visualsonics, Toronto, Canada). The readings of abdominal aortic diameter over three cardiac cycles were averaged and considered as baseline. Reactive hyperemia was induced by inflation of blood pressure cuff, which was modified for mouse use. The cuff was placed on the right lower extremity and kept at 200 mmHg with sphygmomanometer for 2 min. Flow-mediated reactivity was measured 1 min after cuff deflation. After 15 min rest, the mice received sublingually 500 μg nitroglycerin (GTN). The heart rate and diameter of abdominal aorta were measured again after 3 min. For the whole duration of the measurement, the ultrasound probe remained in the same position. EDD and endothelium-independent vasodilation (EID) were calculated as follows: EDD % = $(D_1-D_0)/D_0 \times 100$; EID% = $(D_2-D_0)/D_0 \times 100$. Where $D_1$ is the maximum systolic diameter after FMD and $D_2$ the maximum systolic diameter after nitroglycerin infusion, $D_0$ is the mean diameter of the murine abdominal aorta over three cardiac cycles at baseline. In some cases, 6-week-old WT mice have intraperitoneally injected with vehicle (corn oil) or 2 mg/kg body weight/ day rapamycin (HY-10219, MCE) for 7 consecutive days, which led to a comparable blood concentration of rapamycin in mice to that in patients implanted with DES[17], before transcutaneous ultrasound evaluation of endothelial relaxation function.

**In vivo modulation of endothelial KLF2 or Nox2 expression**. Mouse KLF2-coding or $Cybb$ (i.e., $Nox2$)-targeting sequence was constructed under the EC-specific ICAM2 promoter by Genechem (Shanghai, China) into the adeno-associated virus (AAV, serotype 9) that expresses Flag and enhanced GFP (EGFP). 6-week-old mice were i.v. injected at $3 \times 10^{11}$ virus genomes (vg), followed by

tamoxifen to induce EC-specific depletion of $Rptor$ or $Rictor$, and transcutaneous ultrasound measurement of vasomotor function performed as above.

**Ex vivo measurement of vascular reactivity**. The isometric tension of the mouse aortic rings was measured with $Myograph$ (DMT, Multi Myograph System-620M). Briefly, the mice were sacrificed with overdose of $CO_2$, and their aortas were carefully dissected. The aortas were cleaned of fat and cut into rings approximately 2–3 mm in length, followed by immersing in Krebs solution containing the following components: 120 mmol/L NaCl, 4.7 mmol/L KCl, 1.2 mmol/L MgSO$_4$, 1.18 mmol/L KH$_2$PO$_4$, 24.5 mmol/L NaHCO$_3$, 2.25 mmol/L CaCl$_2$, 11.1 mmol/L glucose, 0.03 mmol/L EDTA-Na$_2$ at pH 7.4 at 37 °C with the gas mixture (95% O$_2$ and 5% CO$_2$) for 30 min for equilibration. The rings were then exposed to KCl (80 mmol/L) until a stable contractile response was reached. After another equilibration, the contraction was induced with $10^{-7}$ mol/L norepinephrine (NE, Sigma-Aldrich) before cumulative vasodilatory response to $10^{-9}-10^{-5}$ mol/L ACh (Sigma-Aldrich) was examined. Relaxation to Ach was expressed as percentage decrease from $10^{-7}$mol/L NE-induced contraction.

**Primary culture of aortic EC**. Briefly, mice were euthanized with an overdose of $CO_2$ before cells were isolated from $Rptor^{EC-/-}$, $Rictor^{EC-/-}$ or $Mtor^{EC-/-}$ mice, and their wild-type littermates using positive immuno-selection with a rat anti-mouse CD31 (BD). EC purity was examined with the uptake of Dil-Ac-LDL (Biomedical Technologies) and FITC-conjugated anti-CD31 (Invitrogen) labeling. Primary purified aortic EC were cultured in EGM2 (Lonza)[50].

**NO measurements**. Both electron paramagnetic resonance (EPR) spectroscopy and a colorimetric assay were applied to measure NO in the murine serum and HAEC culture supernatant.

For ERP measurement, samples were mixed with an equivalent volume of cPTIO (C348, Dojindo Laboratories, Japan) at a final concentration of 0.25 mM and measured in a quartz 1-cm pathlength cuvette at 25 °C. The EPR spectra were recorded by an EPR spectrometry (A300-10/12, Bruker, Germany) with the following instrument settings: modulation frequency, 100 kHz; modulation amplitude, 1 G; sweep width, 100 G; sweep time, 167.772 s; microwave frequency, 9.434 GHz; microwave power, 20.127 mW. As NO trapping causes a characteristic disappearance in the EPR spectrum of cPTIO, the relative NO level was quantified by the decreased EPR signal.

For the colorimetric assay, a commercially available kit (A012-1, Nanjing Jiancheng Bioengineering Institute) was used following the manufacturer's procedure. Briefly, the optical density (OD) values at 550 nm were measured with an ELx800 (BioTek Instruments) after a chromogenic reaction following nitrate reductase-catalyzed conversion of nitrate to nitrite. NO amount was calculated by comparison with a standard and normalized by protein quantity of HAEC in each well of the plate.

**Measurement of superoxide in the mouse serum and HAEC**. $O_2^-\cdot$ in the mouse serum was evaluated with the chemiluminescent dye 9,9′-Bis (N-methylacridinium nitrate (lucigenin, 2315-97-1, Sigma)[51]. Briefly, 15 μl serum was incubated with 85 μl lucigenin (20 μM) for 10 min at 37 °C in dark before measurement with SpectraMax Gemini EM microplate spectrofluorometer (Molecular Devices).

The intracellular $O_2^-\cdot$ was evaluated using a fluorescent dye dihydroethidium (DHE, S0063, Beyotime Institute of Biotechnology)[52]. After treatment, HAEC were harvested, washed twice with PBS, and incubated with 2.5 μM DHE for 30 min at 37 °C in dark. The intracellular accumulation of $O_2^-\cdot$ was then detected with FACSCalibur flow cytometer (BD) followed by analysis with FlowJo.

**Modulation of endothelial KLF2 or Nox2 expression with adeno-associated virus (AAV) delivery approach**. An adeno-associated virus (AAV, serotype 9) expressing mouse $Klf2$ (Gene ID:16598) fused with Flag and enhanced GFP (EGFP) under the endothelial-specific ICAM2 promoter was constructed by and obtained from Genechem (Shanghai, China). Similarly, the same vector containing two tandem sequences targeting mouse $Cybb$ (i.e. $Nox2$, Gene ID:13058) (5'-GCUGCCAGUGUGUCGAAAUTT-3', 5'-CCUAUGUUCCUGUACCUUUTT-3') was obtained.

Six-week-old $Rptor^{EC-/-}$ mice were i.v. injected through the tail vein with $Klf2$-expressing AAV and $Rictor^{EC-/-}$ mice with $Nox2$-interfering AAV at $3 \times 10^{11}$ virus genomes (vg), followed by tamoxifen induction of EC-specific depletion of $Rptor$ or $Rictor$ and transcutaneous ultrasound measurement of vasomotor function as above. The same amount of the empty AAV (AAV-EGFP) was injected as control. After the mice were sacrificed, the frozen section (5 μm) of the aorta was incubated with rat or rabbit anti-mouse CD31 (553370, BD Pharmingen), eNOS (#32027, Cell Signaling Technology) or Nox2 (sc-130543, Santa Cruz Biotechnology) followed by staining with Alexa 594-conjugated goat anti-rat IgG (A11007, Invitrogen) or PE-conjugated mouse anti-rabbit IgG (sc-3753, Santa Cruz Biotechnology) before examined with a fluorescence microscope (Olympus IX53).

**Statistics and reproducibility**. Statistical analyses were performed with GraphPad Prism 7.00 (GraphPad Software, Inc.). Data were presented as mean ± SEM. N

means the independently repeated experiment number. Student's $t$ test or one-sample $t$ test was used to compare the difference between two groups while ANOVA with Dunnett's or Tukey's post hoc correction test was applied to analyze multiple groups. Differences were considered significant at $p \leq 0.05$.

**Reporting summary**. Further information on research design is available in the Nature Research Reporting Summary linked to this article.

## Data availability

The authors declare that all data supporting the findings of this study are available within the paper, Supplementary Information and Supplementary Data 1.

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

## Acknowledgements

We would like to thank Dr. Yang Zhao from the School of Public Health of Nanjing Medical University for valuable advice on statistical analysis, Dr. Yong Ji from Nanjing Medical University for kindly providing Myograph device and professor Jie-Zeng Jiang, Mr. Min-Min Miao, and Ms. Chong Chen from Yangzhou University for valuable help in EPR measurement. This work was supported by the National Natural Science Foundation of China [81870355, 81670410, and 82170465 to C.S. and 81970342 to D.Z.], and Jiangsu Provincial Key Research and Development Program [BE2018611 to D.Z.].

## Author contributions

Design and drafting of the study (C.S., Y.W., and A.G.P.); data acquisition and analysis (Y.W., Q.L., Z.Z., K.P.), data interpretation (Y.W., Q.L., K.P., Z.Z., D.Z., Q.Y., A.G.P., S.I.S., and C.S.).

## Competing interests

The authors declare no competing interests.
