## [Peer Review File · Communications Biology]

Reviewers' comments:

Reviewer #1 (Remarks to the Author):

Although understanding the relationship between mTOR and eNOS leading to changes in NO bioavailability would be of interest to the community, the data presented, the vague methods, and statistical concerns I have limit my ability to make conclusions about the data.

1) rewriting the methods so that key important details for understanding your design are included in the body - not just the supplement.

The methods are way too brief and require a lot of attention. Please move key experimental details into the body because what is written currently is too vague to have any meaning. Your supplement is great, but key details need to be in the body so given word limits be sure you provide the necessary details in the body that can allow anyone to potentially replicate your study.

Line 172 – need more information about at what age, what dose, how many injections, were all mice given tamoxifen? How long was washout?

What sex of mice were used. This needs to be reported and justified.

Please confirm your methods measured NO and not the metabolites nitrite/nitrate? Especially the colorimetric kit. Given NO very fast half-life, why wasn't nitrite also measured in the plasma and cultures?

You state in the results that mice were given injections of rapamycin but the dose is not reported, timing is not reported etc.

You gave mice GTN – how, when, how much

Statistics need more definition. Student's t-test, how many tails, paired/unpaired etc? Why were Dunnett's and Tukey's used as a post hoc (why wasn't it the same post hoc)?

Line 272 – why were Apoe-null mice used? They are not mentioned in the methods and all the details on how you made all your KO's on this background are also missing.

Line 273 – what was the diet? How long were they fed? So many important details that are missing.

2) a lack of rigor and reproducibility in your data presented

A sample size of n=3 is not rigorous for your protein measurements. Why was n=3 for protein used but n= 6 for cDNA?

often your control groups are set to 1 so there is no ability to determine the variability in that group. Thus, your statistics of t-tests is not appropriate because the data are not normally distributed. This makes your interpretation of the data skewed. For example in Figure 5 – there is a lot of variability in your superoxide measurements but yet, because your control groups are 1, you say they are statistically different. Seems highly unlikely in Figure 5B and D (all of them really). Then in Figure 5G you do give the values for the control but yet again there is only 1 point that doesn't overlap with the control group yet you claim they are significantly different.

I think having an n=1 for a western blot, without a quantification is meaningless. Presenting data this way suggests to me that rigor was not taken and there is limited reproducibility in your system. Figure 2 and Figure 4, Figure 5, Figure 6, Figure 7 all of these are n =1 and yet you make conclusions. I don't think this is appropriate.

3) Confirmation of the specificity, % knockdown etc is missing for all the genetically modified mice. Where is the confirmation of each KO line made? This is key and should be presented in the supplement.

Reviewer #2 (Remarks to the Author):

Summary

This manuscript from Wang et al set out to investigate the mechanism through-which drug-eluting stents, a successful therapy in preventing in-stent restenosis, results in the delayed arterial healing and poor reendothelialization in patients. The stents elute allosteric inhibitors to mTOR, which have previously been shown to promote vascular dysfunction. Hence the aim of the authors was to investigate "the mechanisms by which mTOR signaling affects endothelial-dependent vasodilatation (EDD)"

The authors state that pharmacological and genetic inhibition of mTORC1 alone, decreased transcription and expression of eNOS, due to decrease in transcription factor KLF2, and this resulted in the decreased NO availability which gave rise to the observed impaired EDD. While pharmacological and genetic inhibition of mTORC2 alone increased NOX2 via p38/JNK, which increased ROS levels and resulted in feed-forward cycle of eNOS uncoupling and more superoxide production.

General comments

Overall data presented in this manuscript is of value to the scientific community and the wider field. I highly commend the experiments carried out particularly the specificity of animal models and clarity of the figures presented. The true novelty of this study is in attempting to isolate differential contribution of mTORC1 and mTORC2 to endothelial cell dysfunction. However, there are some conceptual concerns to be addressed.

1. There appears to be an oversight in the novelty of the study. The authors attempted to contextualize their data within the field but omitted more recent and key publications (listed below) and hence the novelty emphasis was misplaced. A more thorough review of the literature is recommended, particularly mTOR regulation of vascular function.
2. The authors state "this is the first in vivo mouse study to demonstrate the causative role for mTOR signaling in flow-mediated vasomotor function", however this study by {Reho, 2021 #177@@author-year} was published earlier this year. There are differences in the results obtained by both groups e.g. {Reho, 2021 #177@@author-year} observed no change in eNOS expression which conflicts with data presented in the current manuscript- this ought to be addressed by the authors.
3. The reason behind this study are DES, which are utilized as a potential therapy for stenosis caused by atherosclerotic plaques- hence it was very exciting to see the mTOR/ApoE-/- murine model, it was however disappointing to see they were not fully utilized or discussed. The authors mention an ongoing study with this model, which is encouraging, nevertheless it then begs the question of relevance in the current work. It is a major leap to state exacerbated depression of NO levels as "illustrating the putative importance of impaired mTOR signaling in the presence of dyslipidemia that promotes atherosclerosis". That data is useful for characterizing the significance of mTOR signaling in atherosclerosis with additional data to corroborate the statement. However, its place in the current manuscript is unclear as it does not further elucidate how "mTOR signaling exerts its effect on eNOS production and function in arterial EC".
4. Data presented in the manuscript focuses on potential mechanisms underlying the known mTOR regulation of EDD and eNOS expression/activation-which was described in the introduction. However, this is not fully aligned with the current title. From the current title it is expected that there would be experiments illustrating changes in EDD with deletion of mTOR which is then rescued by over-expression mTOR and similarly with eNOS expression and evidence of eNOS uncoupling.
5. The amount of Western blot data presented is remarkable and are clean, unfortunately in some instances, the differences claimed were not so easy to visualize e.g., Fig. 2F. These blots are vital evidence for the study so would significantly benefit from corresponding graphs. At a minimum quantification of key targets that support/refute mechanisms proposed by authors.
6. Data from Fig.3 is exciting in showing downregulation of eNOS expression. These experiments were done in HAECs, it is unclear what n describes. Is it a description of successful transfections, cell lysates or technical replicates?
7. In Supplementary Methods the authors state: "After pretreatment with chemicals, HAEC were

treated with mTOR inhibitors rapamycin" it is unclear what chemicals are being referred to here.

8. The conclusion from the text and Fig. 5E is that eNOS uncoupling contributes to mTORC2-inhibited ROS production. However, L-NAME was the only NOS inhibitor used, it is a non-selective NOS inhibitor. The authors need to show that mTORC2 inhibition at this dose, does not result in an upregulated iNOS compensation. Additionally, the study would be further strengthened by data showing levels of BH4 and/or eNOS monomer: dimer ratio before a conclusion can be made that the increased mTORC2-inhibition induced ROS levels are solely due to eNOS uncoupling.

9. In line with comments #1 and #4, the protein band of IRF-1 (Fig. 4A) following rapamycin treatment looks darker than control- this does need to be quantified because, IRF-1 regulates transcription of Interferon gamma, which induces iNOS expression. This information does not refute the contribution of eNOS but is more informative in elucidating mechanisms involved.

10. mTOR signaling is a central point in numerous metabolic physiological and pathological pathways. A study by {Decker, 2018 #175@@author-year} similarly looked into mTOR regulation of eNOS phosphorylation- it would be helpful to have a schematic illustrating how data from the current manuscript fits into the field of mTOR-eNOS signaling axis.

11. The statistical analyses utilized in the study seem appropriate for the comparisons being made, with the exception of Figure 6B-D. Comparisons seem to be made between various inhibitors in the presence and absence of torin, a 2-WAY ANOVA seems more appropriate because there are 2 variables- torin and other inhibitors.

Minor comments

- a. There are a couple of typographical errors in supplemental methods and discussion
- b. Section 3.2 second sentence- rewording needed
- c. Section 3.4- "protein expression" is missing in Fig.4A description- way it reads currently contradicts Fig.4f description

Response to reviewers

We thank each reviewer for the positive comments and thorough review of the manuscript. We have carefully addressed each question or concern through additional experiments, improved data presentation, and an expanded discussion. We have transferred more methods details from the supplement to the main text. We have measured BH4 levels using an ELISA assay and evaluated eNOS dimer to monomer ratio with native PAGE followed by Western blot to confirm eNOS uncoupling in mTORC2-inhibited cells (revised Fig 5). We have also examined the effect of mTOR inhibition on iNOS expression (revised Supplemental Figure III-D). Moreover, we have expanded the discussion by citing two relevant reports published recently (Ref #29 and 30). We have also consulted a statistician to improve data presentation and analysis. A schematic diagram has been added (Fig 8). In addition, to avoid possible color page charge, we put EPR data in the Supplemental file. All changes have been highlighted. Please see our point-to-point response below.

Reviewer #1:

rewriting the methods so that key important details for understanding your design are included in the body - not just the supplement. The methods are way too brief and require a lot of attention. Please move key experimental details into the body because what is written currently is too vague to have any meaning. Your supplement is great, but key details need to be in the body so given word limits be sure you provide the necessary details in the body that can allow anyone to potentially replicate your study.

Response: We thank the reviewer for the positive comments on the supplement. As requested, we have added as many key experimental details as possible into the main text within the word limits.

Line 172 – need more information about at what age, what dose, how many injections, were all mice given tamoxifen? How long was washout?

Response: As described in the Supplement, 6-week old mice received intraperitoneal injections of tamoxifen (135mg/kg, dissolved in corn oil) once per day for 7 consecutive days. All mice were given tamoxifen. Washout was two weeks. As requested, we have added these details into page 18 of the main text.

What sex of mice were used. This needs to be reported and justified.

Response: Both male and female mice were used in *in vivo* and *ex vivo* vascular function studies in a sex-matched manner. We have clarified it on page 18. To evaluate the effects of mTOR inhibition in the pro-atherosclerotic settings, only the male mice were used as is typical of atherosclerosis studies. We described this in Supplement.

Please confirm your methods measured NO and not the metabolites nitrite/nitrate? Especially the colorimetric kit. Given NO very fast half-life, why wasn't nitrite also measured in the plasma and cultures?

Response: We completely agree with the reviewer that NO has a very short half-life

and is readily converted to nitrite/nitrate. However, measurement of nitrite/nitrate in the blood or cell culture medium is still a widely used method to indirectly evaluate the production of NO. As reported previously (*Circulation* 1995;91:2982-2988), nitrate/nitrite is a reliable indication of NO formation *in vivo* except for disease states, such as heart failure, in which renal function and extracellular volume are seriously altered. Moreover, in this study we also used electron paramagnetic resonance spectroscopy (EPR) which directly measures NO metabolism.

You state in the results that mice were given injections of rapamycin but the dose is not reported, timing is not reported etc.

Response: The mice received 2mg/ kg body weight/ day rapamycin (HY-10219, MCE) for 7 consecutive days. We have added the information to page 19 of main text.

You gave mice GTN – how, when, how much

Response: 15 min after measurement of flow-mediated reactivity, the mice received sublingually 500µg nitroglycerin (GTN). We have added the information on page 6 of main text.

Statistics need more definition. Student's t-test, how many tails, paired/unpaired etc? Why were Dunnett's and Tukey's used as a post hoc (why wasn't it the same post hoc?)?

Response: All t-test used in this study are two-tailed t-test. We have clarified these in the figure legends. Dunnett's or Tukey's post hoc was used depending on comparison among all the experimental groups or only between experimental groups and the control group.

Line 272 – why were Apoe-null mice used? They are not mentioned in the methods and all the details on how you made all your KO's on this background are also missing.

Response: High-fat diet fed Apoe-null mice are a common animal model of atherosclerosis. As reduced NO bioavailability sets the stage for the initiation of atherosclerosis, ApoE deficiency was further introduced to generate *Apoe^{-/-}Mtor^{EC-/-}*, *Apoe^{-/-}Rptor^{EC-/-}* and *Apoe^{-/-}Rictor^{EC-/-}* mice, which were fed with a high-fat diet for 12 weeks to evaluate the effects of mTOR inhibition on atherosclerosis in our ongoing study. We report the NO data here to demonstrate that the effect of mTOR inhibition on NO production was exacerbated in the pro-atherosclerotic setting. As requested, we have added the information to main text on page 7 that "Inhibition of endothelial mTOR causes impaired EDD, which is ... exacerbated in the pro-atherosclerotic settings where DES is often utilized as a therapy". Moreover, we have discussed on page 14 that "Deficiency of mTOR, Raptor or Rictor exacerbated the NO reduction in the Apoe-null mice, illustrating the putative importance of impaired mTOR signaling in the presence of dyslipidemia that promotes atherosclerosis. Inhibition of mTOR or disruption of either complex in EC may further promote the development of atherosclerosis, which is the subject of our ongoing study." Please also refer to

reviewer #2's question #3.

Line273 – what was the diet? How long were they fed? So many important details that are missing.

Response: Our group always uses Research Diet (D12108C) containing 1.25% cholesterol to feed Apoe-null mice to induce atherosclerosis, which we have added on page 19 of main text and page 2 of the Supplement.

2) a lack of rigor and reproducibility in your data presented

A sample size of n=3 is not rigorous for your protein measurements. Why was n=3 for protein used but n= 6 for cDNA?

Response: We found there is less variation in Western blot study. Usually ≥ 3 independent experiments reached statistical significance for $\alpha=0.05$. For cDNA measurement, however, the variation was higher. Without excluding any data, we had to increase sample size to reach statistical significance.

often your control groups are set to 1 so there is no ability to determine the variability in that group. Thus, your statistics of t-tests is not appropriate because the data are not normally distributed. This makes your interpretation of the data skewed. For example in Figure 5 – there is a lot of variability in your superoxide measurements but yet, because your control groups are 1, you say they are statistically different. Seems highly unlikely in Figure 5B and D (all of them really). Then in Figure 5G you do give the values for the control but yet again there is only 1 point that doesn't overlap with the control group yet you claim they are significantly different.

Response: As there is variation even for the same treatment among different experiments, we presented data from each independent experiment as percentage or fold of its respective control. Therefore, the control group was normalized to "1" or "100". For original Fig 5B and D, we used paired t test. For original Fig 5G (revised Fig 5I), we used repeated measures ANOVA. To address the reviewer's concern, we consulted a statistician (refer to Acknowledgement) who confirmed the suitability of repeated measures ANOVA for ≥ 3 groups and advised to use one-sample t-test for two groups. As paired t test and one-sample t-test resulted in the same p value, it did not change the conclusions. We have clarified the statistical analysis in each figure legend.

I think having an n=1 for a western blot, without a quantification is meaningless. Presenting data this way suggests to me that rigor was not taken and there is limited reproducibility in your system. Figure 2 and Figure 4, Figure 5, Figure 6, Figure 7 all of these are n =1 and yet you make conclusions. I don't think this is appropriate.

Response: The reviewer misunderstood. Due to the limitation of the space of each figure, we showed the quantification data of the most important Western blot results. For others we only showed the representative images. As we described in the

Supplement, all Western blot images shown in this study were representative images from at least three independent experiments.

3) Confirmation of the specificity, % knockdown etc is missing for all the genetically modified mice. Where is the confirmation of each KO line made? This is key and should be presented in the supplement.

Response: Thank you. When we were submitting this manuscript, another study of our group using the same mice was under revision. Therefore, we did not include the data that validated the knockout efficiency in this submission. As that manuscript is now published, we have cited it (Ref #18, *Am J Respir Cell Mol Biol.* 2021 Dec;65(6):646-657). For the reviewer's convenience, we have copied the relevant validation data below.

Figure E2. Representative Western blotting of EC isolated from *Mtor*^{EC-/-} (A), *Raptor*^{EC-/-} (D), *Rictor*^{EC-/-} (E) and the respective WT mice. * $p < 0.05$, ** $p < 0.01$, two-tailed t-test.

Reviewer #2:

1. There appears to be an oversight in the novelty of the study. The authors attempted to contextualize their data within the field but omitted more recent and key publications (listed below) and hence the novelty emphasis was misplaced. A more thorough review of the literature is recommended, particularly mTOR regulation of vascular function.

Response: We sincerely apologize for incomplete review of relevant publications from 2021 in our original submission as we prepared our manuscript in middle 2020. As kindly recommended by the reviewer, we have carefully read and cited two publications from the same group (Ref #29: *Hypertension*. 77: 594–604, 2021; and Ref #30: *Am J Physiol Regul Integr Comp Physiol*. 321: R228–R237, 2021).

In *Am J Physiol Regul Integr Comp Physiol* study, the authors report increased vascular mTORC1 signaling in obesity. However, deletion of endothelial Raptor (critical subunit of mTORC1) failed to reverse the endothelial dysfunction caused by a high-fat/high-sucrose diet in mice. In the *Hypertension* study, the authors further report that endothelial cell-specific Raptor deletion results in reduced relaxation responses evoked by acetylcholine in the aorta, which was reversed by restoration of mTORC1 signaling through overexpression of p70S6K in aortic rings. Both studies mainly applied *ex vivo* experiments in which the aortic and mesenteric rings were used to examine the effect of endothelial Raptor deficiency on vascular function. Our manuscript differs from these studies in that *in vivo* and *ex vivo* results were complemented by a large number of cell culture studies to dissect the molecular mechanisms underlying the clinical phenomena that application of mTOR inhibition therapeutics is associated with high risk of vascular dysfunction. In addition to endothelial Raptor-deficient mice, we also examined the vascular function of endothelial mTOR- and Rictor-deleted mice. Combining the data from cell culture, we thoroughly revealed two synergistic mechanisms through which inhibition of mTOR complex 1 and/or 2 impaired endothelium-dependent vasorelaxation. Therefore, our manuscript extends the previous studies with novel mechanistic contributions.

2. The authors state “this is the first *in vivo* mouse study to demonstrate the causative role for mTOR signaling in flow-mediated vasomotor function”, however this study by {Reho, 2021 #177@@author-year} was published earlier this year. There are differences in the results obtained by both groups e.g. {Reho, 2021 #177@@author-year} observed no change in eNOS expression which conflicts with data presented in the current manuscript- this ought to be addressed by the authors.

Response: 1) Neither of Reho’s two publications mentioned above studied flow-mediated vasomotor function *in vivo*. Therefore, it is true that our manuscript is the first to apply *in vivo* non-invasive transcutaneous ultrasound imaging to demonstrate the causative role for mTOR signaling in flow-mediated vasomotor function. 2) Yes, there are differences in the results of Reho’s two studies (see above) and our manuscript is in line with the observation reported in *Hypertension*. 3) As for eNOS expression, both of Reho’s studies examined the whole aortic lysates prepared from endothelial Raptor-

deficient mice. As endothelial cells only constitute one thin layer of the aorta, we do not think any solid conclusion can be drawn from the WB results. At least, those results do not appear to conflict with data presented in our manuscript to any great extent, given differences in experimental details. Please refer to the discussion on page 14-15.

3. The reason behind this study are DES, which are utilized as a potential therapy for stenosis caused by atherosclerotic plaques- hence it was very exciting to see the mTOR/ApoE^{-/-} murine model, it was however disappointing to see they were not fully utilized or discussed. The authors mention an ongoing study with this model, which is encouraging, nevertheless it then begs the question of relevance in the current work. It is a major leap to state exacerbated depression of NO levels as “illustrating the putative importance of impaired mTOR signaling in the presence of dyslipidemia that promotes atherosclerosis”. That data is useful for characterizing the significance of mTOR signaling in atherosclerosis with additional data to corroborate the statement. However, its place in the current manuscript is unclear as it does not further elucidate how “mTOR signaling exerts its effect on eNOS production and function in arterial EC”.

Response: Thank you for your helpful comments. As reduced NO bioavailability set the stage for the initiation of atherosclerosis, we generated *ApoE^{-/-}Mtor^{EC-/-}*, *ApoE^{-/-}Rptor^{EC-/-}* and *ApoE^{-/-}Rictor^{EC-/-}* mice which were fed with a high-fat diet for 12 weeks to evaluate the effects of mTOR inhibition on atherosclerosis, a study which we are wrapping up now. Due to the word and figure limits and the scope of this manuscript, we are going to report the results in separate publication. Only NO data here were shown in this manuscript to demonstrate that the deleterious effect of mTOR inhibition on NO production was aggravated in the pro-atherosclerotic settings. Combining the comment from reviewer #1, we have added the information to main text on page 7 that “Inhibition of endothelial mTOR causes impaired EDD, which is ... exacerbated in the pro-atherosclerotic settings where DES is often utilized as a therapy”. Moreover, we have discussed on page 14 that “Deficiency of mTOR, Raptor or Rictor exacerbated the NO reduction in the *ApoE*-null mice, illustrating the putative importance of impaired mTOR signaling in the presence of dyslipidemia that promotes atherosclerosis. Inhibition of mTOR or disruption of either complex in EC may further promote the development of atherosclerosis, which is the subject of our ongoing study.”

4. Data presented in the manuscript focuses on potential mechanisms underlying the known mTOR regulation of EDD and eNOS expression/activation- which was described in the introduction. However, this is not fully aligned with the current title. From the current title it is expected that there would be experiments illustrating changes in EDD with deletion of mTOR which is then rescued by over-expression mTOR and similarly with eNOS expression and evidence of eNOS uncoupling.

Response: Thank you. We did examine EDD change in mTOR deletion. We also examined EDD change in Raptor and Rictor deletion. As we further characterized

inhibited p70S6K/ KLF2 and activated MAPK/Nox2 contributed to impaired EDD in mTORC1 and mTORC2 inhibition respectively, we rescued EDD in endothelial Raptor-deficient mice with KLF2 overexpression and rescued endothelial Rictor-deficient mice with Nox2 knockdown. Compared with simple overexpression of mTOR, we believe our strategy is more specific and that the title is appropriate.

5. The amount of Western blot data presented is remarkable and are clean, unfortunately in some instances, the differences claimed were not so easy to visualize e.g., Fig. 2F. These blots are vital evidence for the study so would significantly benefit from corresponding graphs. At a minimum quantification of key targets that support/refute mechanisms proposed by authors.

Response: Thank you. Considering the space, we did not provide quantification for each Western blot but only very important targets (such as eNOS) in our original submission. As recommended, we have provided quantification data also for key targets including KLF2 in revised Fig 4A and 4H; Nox2 in revised Fig 5K-L and Fig 6E; phosphorylation of p38 and JNK in revised Fig 6F. As for Fig. 2F (revised Fig 2D), as very reliable commercial siRNAs have been used and all the signal molecules examined are known to be downstream of mTOR signaling, we believe they are not key targets and thus only representative images provided.

6. Data from Fig.3 is exciting in showing downregulation of eNOS expression. These experiments were done in HAECs, it is unclear what n describes. Is it a description of successful transfections, cell lysates or technical replicates?

Response: Thank you. N means the independently repeated experiment number, so is a form of technical replication.

7. In Supplementary Methods the authors state: “After pretreatment with chemicals, HAEC were treated with mTOR inhibitors rapamycin” it is unclear what chemicals are being referred to here.

Response: Thank you. Here “chemicals” mean pharmacological inhibitors used in this study, such as NOS inhibitor L-NAME, PKC δ inhibitor Rottlerin, PKC α and PKC β I inhibitor Gö-6976, ROS scavenger N-acetylcysteine, mitochondrial antioxidant MitoQ, Nox inhibitor Apocynin, XO inhibitor Allopurinol, p38 inhibitor SB 203580, JNK inhibitor SP 600125 and ERK inhibitor U0126, etc. For each pretreatment, please refer to the figure legends.

8. The conclusion from the text and Fig. 5E is that eNOS uncoupling contributes to mTORC2-inhibited ROS production. However, L-NAME was the only NOS inhibitor used, it is a non-selective NOS inhibitor. The authors need to show that mTORC2 inhibition at this dose, does not result in an upregulated iNOS compensation. Additionally, the study would be further strengthened by data showing levels of BH4 and/or eNOS monomer: dimer ratio before a conclusion can be made that the increased mTORC2-inhibition induced ROS levels are solely due to eNOS uncoupling.

Response: Thank you for these helpful suggestions. 1) We have purchased an anti-iNOS antibody (ab178945, Abcam) and probed for iNOS expression in torin-treated HAEC with Western blot. At the dose and time used in this study, iNOS expression was not significantly changed by mTORC2 inhibition. The result has been added to the revised Supplemental Figure III-D and described on page 10 of main text. 2) Using ELISA, we have measured BH4 level in mTOR-inhibited HAEC. The results indicated that torin but not rapamycin treatment decreased BH4 level (revised Figure 5E). 3) We have also applied native PAGE and analyzed the relative amount of eNOS monomer and dimer. The results revealed torin 1 but not rapamycin treatment increased the monomer to dimer ratio (i.e. decreased dimer to monomer ratio) (revised Fig 5F).

9. In line with comments #1 and #4 the protein band of IRF-1 (Fig. 4A) following rapamycin treatment looks darker than control- this does need to be quantified because, IRF-1 regulates transcription of Interferon gamma, which induces iNOS expression. This information does not refute the contribution of eNOS but is more informative in elucidating mechanisms involved.

Response: The reviewer is correct that IRF-1 expression was upregulated by rapamycin treatment, a result which we reported in our previous study (Ref #20: *J Mol Cell Cardiol* 140, 30-41 (2020)). We have cited this publication and described this result on page 9. Moreover, it is true that IRF1 was widely reported to regulate iNOS expression in macrophages (PLoS One. 2015 Feb 6;10(2):e0117782. *J Biol Chem.* 2003 Jan 24;278(4):2271-7.). Several studies also indicated that IRF1 regulated iNOS expression and NO production in rodent SMC (*Am J Physiol Cell Physiol.* 2002 Jan;282(1):C144-52) and EC (*Cancer Res.* 2005 Sep 1;65(17):7984-92; *Cell Mol Life Sci.* 2003 Mar;60(3):518-25.) and Human venous EC (*Life Sci.* 2019 Sep 15;233:116525). However, our results indicated that iNOS expression in HAEC was not significantly affected by rapamycin or torin. A possible reason is the short treatment time. We used 1h, which is probably not long enough for IRF1 to bind to the element located on the iNOS gene promoter although IRF1 expression is upregulated. Another possible reason is IRF-1 alone is not enough to upregulate iNOS expression, as it was reported in macrophage that a complex formation of ICSBP with IRF-1 is essential for iNOS expression and NO production (*J Biol Chem.* 2003 Jan 24;278(4):2271-7).

10. mTOR signaling is a central point in numerous metabolic physiological and pathological pathways. A study by {Decker, 2018 #175@@author-year} similarly looked into mTOR regulation of eNOS phosphorylation- it would be helpful to have a schematic illustrating how data from the current manuscript fits into the field of mTOR-eNOS signaling axis.

Response: Thank you. A schematic diagram has been added (Fig 8).

11. The statistical analyses utilized in the study seem appropriate for the comparisons being made, with the exception of Figure 6B-D. Comparisons seem to be made between various inhibitors in the presence and absence of torin, a 2-WAY ANOVA seems more appropriate because there are 2 variables- torin and

other inhibitors.

Response: Thank you very much. We have separated the data without and with torin treatment. Moreover, we have performed 3 additional experiments for revised Fig 6B. Please refer to the revised Fig 6 and the legend.

Reviewers' comments:

Reviewer #1 (Remarks to the Author):

Thank you for your revision,

I would still like to see all of your western blots quantified and individual samples plotted in the supplement and not just the ones you think are the most important. Without this, it really doesn't matter your sample size because readers only get to see 1 representative gel instead of $n = 3$ so they can decide themselves if it is "representative". Again speaks to the rigor and reproducibility of your science.

Reviewer #2 (Remarks to the Author):

The authors did a remarkable job of responding to both reviewers' comments especially with respect to additional information on the ApoE^{-/-} model and iNOS expression experiments. It was a pleasure to read through the manuscript again with these additions and to see detailed research and thought went into the authors' responses.

Minor comments

1.) The schematic in Figure 8 is very helpful in highlighting the key-players involved, however the colors of the arrows in the schematic don't seem to reflect the figure legend or dataset. E.g., both sides result in inhibited NO bioavailability (red- line bar); Torin-1 inhibition of mTORC2 resulted in p38 and JNK phosphorylation (black-straight arrows); Inhibition of mTORC1 impaired p70S6K phosphorylation and KLF2 (red- line bar).

2.) (Figure 10-Q in the online-only Data Supplement) ... should be Figure I-O-Q in the online-only Data Supplement)

Response to Reviewers

Reviewer #1 (Remarks to the Author):

Thank you for your revision, I would still like to see all of your western blots quantified and individual samples plotted in the supplement and not just the ones you think are the most important. Without this, it really doesn't matter your sample size because readers only get to see 1 representative gel instead of $n = 3$ so they can decide themselves if it is "representative". Again speaks to the rigor and reproducibility of your science.

Response: We totally agree with the reviewer in the emphasis on rigor and reproducibility. As requested, we have attached our original Western blot images. Please understand that we usually cut the Western blot membrane into 2-4 pieces (based on the pre-stained marker) in order to probe several proteins in one experiment, after we have tested each antibody for its specificity and target protein size. Here we have made the effort to find and stitch the individual pieces back together.

Reviewer #2 (Remarks to the Author):

The authors did a remarkable job of responding to both reviewers' comments especially with respect to additional information on the ApoE^{-/-} model and iNOS expression experiments. It was a pleasure to read through the manuscript again with these additions and to see detailed research and thought went into the authors' responses.

Response: We thank the reviewer very much for the positive comments on our revision.

Minor comments

1.) The schematic in Figure 8 is very helpful in highlighting the key-players involved, however the colors of the arrows in the schematic don't seem to reflect the figure legend or dataset. E.g., both sides result in inhibited NO bioavailability (red- line bar); Torin-1 inhibition of mTORC2 resulted in p38 and JNK phosphorylation (black-straight arrows); Inhibition of mTORC1 impaired p70S6K phosphorylation and KLF2 (red- line bar).

Response: The reviewer is correct that inhibition of either mTORC1 (left side) or mTORC2 (right side), using rapamycin and/ or torin 1 as shown, ultimately results in inhibited NO bioavailability by perturbing the illustrated mechanisms. On the left side, inhibition of mTORC1 suppresses the illustrated pathway leading to eNOS downregulation through inhibiting p70S6K phosphorylation and KLF2 expression. On the right side, inhibition of mTORC2 enables eNOS uncoupling through activation of p38 and JNK (normally suppressed) and accumulation of ROS. As illustrated, eNOS expression and eNOS uncoupling have opposing effects on NO bioavailability, which is overall enhanced by normal signaling and suppressed by mTOR inhibition. Therefore, the colors of the arrows in the figure reflect normal mTOR signaling, which is perturbed by disruption of either complex (step 1). We believe this would constitute a standard way of illustrating the phenomenon, and that it is consistent with the data in the manuscript. We have also made changes in the legend for clarity.

2.) (Figure 1O-Q in the online-only Data Supplement) ... should be Figure I-O-Q in the online-only Data Supplement)

Response: Thank you very much for your careful review. We have corrected it.